# Four Unity/Variable Gain First-Order Cascaded Voltage-Mode All-Pass Filters and Their Fully Uncoupled Quadrature Sinusoidal Oscillator Applications

**DOI:** 10.3390/s22166250

**Published:** 2022-08-19

**Authors:** Hua-Pin Chen, San-Fu Wang, Yitsen Ku, Yuan-Cheng Yi, Yi-Fang Li, Yu-Hsi Chen

**Affiliations:** 1Department of Electronic Engineering, Ming Chi University of Technology, New Taipei 24301, Taiwan; 2Department of Electronic Engineering, National Chin-Yi University of Technology, Taichung 41170, Taiwan; 3Department of Electrical Engineering, California State University Fullerton, Fullerton, CA 92831, USA; 4Department of Electrical Engineering, National Formosa University, Huwei 63201, Taiwan

**Keywords:** first-order filter, oscillator, active circuit, voltage-mode filter, current conveyor

## Abstract

This paper presents four new designs for a first-order voltage-mode (VM) all-pass filter (APF) circuit based on two single-output positive differential voltage current conveyors (DVCCs). The first two proposed VMAPFs with unity-gain, high-input (HI) impedance and low-output (LO) impedance use two DVCCs, a grounded capacitor, and a grounded resistor. The last two proposed first-order VMAPFs with HI impedance and variable-gain control are two resistors added to each of the first two VMAPFs. The last two proposed first-order VMAPFs with variable-gain control use two DVCCs, one grounded capacitor, and three grounded resistors and provide HI impedances, so that VMAPFs can be directly cascaded to obtain high-order filters without additional voltage buffers. The four implementation circuits based only on grounded passive components are particularly applicable for integrated circuits (ICs). To confirm the cascading characteristics, an application example of a fully-uncoupled quadrature sinusoidal oscillator (FQSO) is also proposed. PSpice simulation results have confirmed the feasibility of the proposed structures. VMAPF and FQSO circuits are also constructed from commercial AD8130 and AD844 ICs, and their experimentally measured time and frequency responses are compared to theoretical values. The supply voltages for both the AD8130 and AD844 ICs were ±5 V. The measured power dissipation of the proposed first-order VMAPF and second-order FQSO circuits is 0.6 W. The measured input 1-dB compression point for the four VMAPFs is about 19 dB. The measured total harmonic distortion of the four VMAPFs is less than 0.67% when the input voltage reaches 2.5 V_pp_. The calculated figures of merit for the four VMAPFs are 628.2 × 10^3^, 603.06 × 10^3^, 516.53 × 10^3^, and 496.42 × 10^3^, respectively.

## 1. Introduction

Voltage-mode (VM) all-pass filters (APFs) are one of the most important components in many sensors, communications, test circuits, and analog signal generation/processing circuits, producing a 180° phase shift and acting as a phase corrector [1,2,3,4,5]. In addition, a quadrature oscillator can be easily implemented by cascading an inverting VMAPF circuit and a non-inverting VMAPF circuit [6]. Many VMAPF circuits have been created in the literature using a single active building block [7,8,9,10,11,12,13,14,15,16,17,18,19,20] or two active building blocks [21,22,23,24]. The single active component reported in [7,8,9,10,11,12,13,14,15,16,17,18,19,20] can provide VMAPF circuits, but they require passive component matching conditions to achieve VMAPF circuits. VMAPF circuits based on two active components have also been reported in [21,22,23,24], but they do not offer variable-gain control. Four HI impedance VMAPF circuits based on fully differential current conveyor (FDCCII) have been proposed in [25,26], but they still do not offer variable-gain control. In addition, the port relationships of the FDCCII are arithmetically equivalent to that of two differential difference current conveyor (DDCC) configurations. DDCC-based VMAPF circuits were proposed in [27,28,29,30,31,32], but they do not provide cascadable fully-uncoupled quadrature sinusoidal oscillator (FQSO) characteristics without adding additional different circuit structures. The circuits proposed in [33,34,35] have variable-gain control capability, but they require passive component matching conditions to achieve VMAPFs. Some single-output positive and negative differential voltage current conveyor (DVCC+/DVCC−) were reported in [36,37,38,39,40], but they do not offer variable-gain control. To avoid loading issues when cascading VMAPFs into larger systems or to make it easier to connect within a system without additional voltage buffers, VM circuits need HI and LO impedance characteristics. In 2022, two VMAPFs based on single-modified negative DDCC (DDCC−) were proposed in [41]. The first proposed VMAPF in [41] uses a modified DDCC−, two grounded capacitors, and a floating resistor, while the second one uses a grounded capacitor, a grounded resistor, and a floating resistor. Both proposed VMAPFs have HI impedance, but they require passive component matching conditions to achieve the VMAPFs and do not offer variable-gain control and LI impedance. In this paper, four new first-order cascadable VMAPFs with unity/variable-gain control are proposed. The first two VMAPFs employ two DVCCs, one grounded capacitor (GC), and one grounded resistor (GR), while the other two VMAPFs employ two DVCCs, one GC, and three GRs. Positive DVCC configuration is simpler than the dual/negative-type DVCC configuration or the modified DDCC− configuration. Both the first two proposed VMAPFs have HI and LO impedances. Two additional first-order VMAPFs with variable-gain control are also demonstrated by adding two GRs to each of the first two VMAPFs. In addition, the application of cascading two variable-gain control VMAPFs to realize FQSO is also given. An overview of the proposed four VMAPF circuits compared to previously reported circuits is shown in Table 1, which compares their use of active and passive elements, use of only grounded passive elements, unconstrained elements matching, HI impedance, variable-gain control, LO impedance, and cascadable synthesis FQSO characteristics without adding additional different circuit structures. In addition to using only grounded passive and unconstrained element matching, the first two circuits have HI and LO impedances, while the latter two circuits have HI impedance and variable-gain control. Furthermore, the last two circuits discussed in this paper for the variable-gain control VMAPFs can be cascaded to synthesize the FQSO without adding another different circuit structure.

## 2. Circuit Description

### 2.1. Basic Concept of Non-Inverting and Inverting VMAPF Functions

One way to implement the non-inverting and inverting VMAPF functions is to generate the difference functions V_o_(s) = V_in_(s) − 2V_1_(s) and V_o_(s) = 2V_1_(s) − V_in_(s), respectively, where V_o_(s) and V_in_(s) are the output and input voltages, and V_1_(s) is the internal node voltage. If the node voltage V_1_(s) realizes the non-inverting first-order low-pass filtering function with the minimum resistor R and capacitor C
(1)$$V_{1}(s)=\frac{\frac{1}{\mathrm{RC}}}{s+\frac{1}{\mathrm{RC}}}V_{\mathrm{in}}(s)$$
then the form of the non-inverting and inverting VMAPF functions can be realized as follows:(2)$$\text{Non-inverting VMAPF}:\;\frac{V_{o}(s)}{V_{\mathrm{in}}(s)}=\frac{s-\frac{1}{\mathrm{RC}}}{s+\frac{1}{\mathrm{RC}}}$$
(3)$$\mathrm{Inverting}\;\mathrm{VMAPF}:\;\frac{V_{o}(s)}{V_{\mathrm{in}}(s)}=\frac{-(s-\frac{1}{\mathrm{RC}})}{s+\frac{1}{\mathrm{RC}}}$$

In Equations (1)–(3), the product RC is the time constant of the non-inverting and inverting VMAPF functions. Based on Equations (2) and (3), when the frequency domain s = jω, and ω is the angular frequency, the gain and phase responses are given by
(4)$$\text{Non-inverting VMAPF}:\;\left|\frac{V_{o}(j\omega )}{V_{\mathrm{in}}(j\omega )}\right|=1,\;∠\varphi_{n}=\pi -2\arctan(\omega \mathrm{RC})$$
(5)$$\mathrm{Inverting}\;\mathrm{VMAPF}:\;\left|\frac{V_{o}(j\omega )}{V_{\mathrm{in}}(j\omega )}\right|=1,\;∠\varphi_{i}=-2\arctan(\omega \mathrm{RC})$$

Based on Equations (4) and (5), the VMAPF functions maintain a constant gain while shifting the phase of the input signal. The phase characteristic in Equation (4) provides a phase shift between 180° and 0° as the input frequency increases and has a phase shift value of 90° at the pole angular frequency of ω_o_. The phase characteristic in Equation (5) provides a phase shift between 0° and −180° as the input frequency increases and has a phase shift value of −90° at the pole angular frequency of ω_o_. Hence, the non-inverting VMAPF characteristic of the phase shift is a leading phase shifter, and the inverting VMAPF characteristic of the phase shift is a lagging phase shifter. The pole angular frequency ω_o_ of the non-inverting and inverting AP filtering functions can be expressed as
(6)$$\omega_{o}=\frac{1}{\mathrm{RC}}$$

### 2.2. Proposed Four VMAPF Circuits and Application Example

The circuit symbol of the DVCC+ and its behavioral model are shown in Figure 1a and Figure 1b, respectively. DVCC+ can be used for a differential voltage and is a variant of current conveyor with a low impedance current input port X, a high impedance current output port Z+, and two high impedance voltage input ports Y_1_ and Y_2_.

DVCC+ with differential input impedances is suitable for processing differential signals because it has two HI impedance terminals. The port relationships of the single-output DVCC+ is characterized by the following matrix [42]:(7)$$\left[\begin{array}{c} I_{Y1}\\ I_{Y2}\\ V_{X}\\ I_{Z} \end{array}\right]=\left[\begin{array}{ccc} 0 & 0 & 0\\ 0 & 0 & 0\\ 1 & -1 & 0\\ 0 & 0 & 1 \end{array}\right]\left[\begin{array}{c} V_{Y1}\\ V_{Y2}\\ I_{X} \end{array}\right]$$

According to Equation (7), the voltage Vx = V_Y1_ − V_Y2_ means that the voltage on port X is the differential sensor voltage for the input ports Y_1_ and Y_2_. The current I_Z_ = I_X_ means that the output current of port Z+ equals the input current of port X. The proposed circuits utilize two DVCCs and two/four grounded passive components. Figure 2a and Figure 2b show the first two proposed VMAPF circuits, respectively, with HI impedance at the input voltage terminal and LO impedance at the output voltage terminal. In Figure 2a,b, the symbol V_in_ is the input voltage; the symbol V_1_ is the internal node voltage; the symbols V_o1_ and V_o2_ are the output voltages, and the symbols C_1_, C_2_, R_1_, and R_4_ are the capacitors and resistors. Each proposed VMAPF employs two DVCCs, one GC, and one GR, which can be directly cascaded for a higher-order filter without additional voltage buffers. Considering the proposed VMAPF in Figure 2a, the nodal equations can be obtained as:
(8)$${(\mathrm{sC}}_{1}R_{1}+{1)V}_{1}=V_{\mathrm{in}}$$
(9)$$V_{o1}=V_{\mathrm{in}}-2V_{1}$$

Solving Equations (8) and (9), the non-inverting APF transfer function can be obtained as:(10)$$\frac{V_{o1}}{V_{\mathrm{in}}}=\frac{s-\frac{1}{C_{1}R_{1}}}{s+\frac{1}{C_{1}R_{1}}}=\frac{s-\omega_{o}}{s+\omega_{o}}$$

Based on Equation (10), the non-inverting APF has a pole angular frequency ω_o_ as follows:(11)$$\omega_{o}=\frac{1}{C_{1}R_{1}}$$

When the capacitor C_1_ = C and resistor R_1_ = R, the gain and phase responses of the non-inverting APF are the same as given in Equation (4).

Considering the proposed VMAPF in Figure 2b, the nodal equations can be obtained as:(12)$${(\mathrm{sC}}_{2}R_{4}+{1)V}_{1}=V_{\mathrm{in}}$$
(13)$$V_{o2}=2V_{1}-V_{\mathrm{in}}$$

Solving Equations (12) and (13), the inverting APF transfer function can be obtained as:(14)$$\frac{V_{o2}}{V_{\mathrm{in}}}=\frac{-(s-\frac{1}{C_{2}R_{4}})}{s+\frac{1}{C_{2}R_{4}}}$$

According to Equation (14), the pole angular frequency of the inverting APF is ω_o_ = 1/C_2_R_4_. When the capacitor C_2_ = C and resistor R_4_ = R, the gain and phase responses of the inverting APF are the same as given in Equation (5).

According to Equations (10) and (14), both the non-inverting APF and the inverting APF have unity-gain. By adding two GRs to the first two VMAPFs in Figure 2a and Figure 2b, respectively, two other first-order VMAPFs with HI impedance and variable-gain control were realized, as shown in Figure 3a and Figure 3b, respectively. In Figure 3a,b, the symbol V_in_ is the input voltage; the symbols V_o3_ and V_o4_ are the output voltages, and the symbols C_1_, C_2_, R_1_, R_2_, R_3_, R_4_, R_5_, and R_6_ are capacitors and resistors.

The analysis of Figure 3a leads to the non-inverting APF transfer function as follows:(15)$$\frac{V_{o3}}{V_{\mathrm{in}}}=(\frac{R_{3}}{R_{2}})(\frac{s-\frac{1}{C_{1}R_{1}}}{s+\frac{1}{C_{1}R_{1}}})=k_{1}(\frac{s-\omega_{1}}{s+\omega_{1}})$$

Based on Equation (15), the non-inverting APF has the following pole angular frequency ω_1_ and variable-gain control k_1_:(16)$$\omega_{1}=\frac{1}{C_{1}R_{1}},\;k_{1}=\frac{R_{3}}{R_{2}}$$

Similarly, the analysis of Figure 3b leads to the inverting APF transfer function as follows:(17)$$\frac{V_{o4}}{V_{\mathrm{in}}}=-(\frac{R_{6}}{R_{5}})(\frac{s-\frac{1}{C_{2}R_{4}}}{s+\frac{1}{C_{2}R_{4}}})=-k_{2}(\frac{s-\omega_{2}}{s+\omega_{2}})$$

Based on Equation (17), the inverting APF has the following pole angular frequency ω_2_ and variable-gain control k_2_:(18)$$\omega_{2}=\frac{1}{C_{2}R_{4}},\;k_{2}=\frac{R_{6}}{R_{5}}$$

Thus, each structure proposed in Figure 3a,b has variable-gain control without affecting the pole angular frequency. Since the circuits of Figure 3a,b have HI impedance, the VM FQSO can be easily implemented by cascading the proposed non-inverting VMAPF and inverting VMAPF circuits without adding additional different circuit structures. To confirm the cascading characterizing of Figure 3a,b, the VM FQSO is implemented using variable-gain control of the proposed non-inverting and inverting APFs. Figure 4 shows the application of the proposed VMAPFs to synthesize a VM FQSO by cascading a non-inverting APF transfer function and an inverting APF transfer function with variable-gain control into the feedback loop.

Based on the cascaded non-inverting and inverting APF transfer functions of Figure 4, the characteristic equation (CE) can be obtained.
(19)$$\mathrm{CE}:\;s^{2}+s(\omega_{1}+\omega_{2})(\frac{1-k_{1}k_{2}}{1+k_{1}k_{2}})+\omega_{1}\omega_{2}=0$$
where $\omega_{1}=\frac{1}{R_{1}C_{1}}$, $\omega_{2}=\frac{1}{R_{4}C_{2}}$, $k_{1}=\frac{R_{3}}{R_{2}}$, and $k_{2}=\frac{R_{6}}{R_{5}}$.

According to Equation (19), the frequency of oscillation (FO) and the condition of oscillation (CO) are
(20)$$\mathrm{FO}:\;f_{o}=\frac{1}{2\pi }\sqrt{\frac{1}{R_{1}R_{4}C_{1}C_{2}}}$$
(21)$$\mathrm{CO}:\;1\leq (\frac{R_{3}}{R_{2}})(\frac{R_{6}}{R_{5}})$$

Examining Equations (20) and (21), FO and CO are fully decoupled. Therefore, FO can be independently adjusted by R_1_ and R_4_, and CO can be independently adjusted by R_2_, R_3_, R_5_, and R_6_. Using the oscillator building blocks in Figure 4, Figure 5 shows how the non-inverting and inverting APFs of Figure 3a,b can be cascaded to synthesize the VM FQSO, CE, FO, and CO derived from Figure 5 are the same as Equations (19), (20), and (21), respectively.

## 3. Simulation Results

To validate the theoretical analyses, simulations were performed using PSpice, and experimental measurements were performed using AD8130 [43,44] and AD844 [45] commercially available components to determine the feasibility and accuracy of the proposed first-order VMAPFs and second-order FQSO. The supply voltages for both the AD8130 and AD844 were ±5 V. Figure 6 shows a possible equivalent implementation using Analog Devices AD8130 and AD844 ICs instead of DVCC+. To obtain f_o_ = 62.41 kHz, a grounded capacitor of 510 pF and a grounded resistance of 5 kΩ were chosen in Figure 2a,b. Figure 7 and Figure 8 show the simulated gain and phase frequency responses of the first two proposed circuits for the unity-gain of the non-inverting VMAPF and the inverting VMAPF in Figure 2a and Figure 2b, respectively. For the non-inverting VMAPF pole frequency with a theoretical phase shift of 90°, the simulated pole frequency in Figure 7 is 60.81 kHz with an offset of −1.6 kHz. For the inverting VMAPF pole frequency with a theoretical phase shift of −90°, the simulated pole frequency in Figure 8 is 60.95 kHz with an offset of −1.46 kHz.

A single grounded capacitor of 510 pF and three equal grounded resistors of 5 kΩ, and the last two proposed variable-gain control non-inverting and inverting VMAPF circuits, shown in Figure 3a,b, were designed for f_o_ = 62.41 kHz. To verify the last two proposed variable-gain control non-inverting and inverting VMAPF circuits in Figure 3a,b, Figure 9 and Figure 10 show the frequency domain simulations of the non-inverting and inverting VMAPFs with an ideal pole frequency of 62.41 kHz, respectively. For a non-inverting VMAPF pole frequency with a theoretical phase shift of 90°, the simulated pole frequency in Figure 9 is 60.3 kHz with an offset of −2.11 kHz. For an inverting VMAPF pole frequency with a theoretical phase shift of −90°, the simulated pole frequency in Figure 10 is 60.5 kHz, and the offset is −1.91 kHz.

## 4. Experimental Results

To measure the gain and phase responses of the first-order VMAPF in the frequency domain, the receiver resolution bandwidth of the Keysight E5061B-3L5 network analyzer was fixed at 100 Hz. To measure circuits in time-domain input/output waveforms, the Tektronix AFG1022 signal generator applied 2.5 V_PP_ to the first-order VMAPF circuits and used an oscilloscope Tektronix DPO 2048B to measure. The Keysight-Agilent N9000A signal analyzer evaluated third-order intermodulation distortion (IMD3), third-order intercept (TOI), phase noise (PN), total harmonic distortion (THD), spurious-free dynamic range (SFDR), and 1-dB compression point (P1dB) analysis. Figure 11 shows photographs of the top and bottom views of the non-inverting and inverting first-order VMAPFs or FQSO printed circuit board (PCB) hardware implementation. Figure 12 shows the photograph of the hardware setup used to experimentally verify the performance of the proposed circuits and shows the gain and phase responses in the frequency of the non-inverting VMAPF as a test case. The supply voltages for both the AD8130 and AD844 are ± 5 V. The measured power dissipation of the proposed first-order VMAPF and second-order FQSO circuits is 0.6 W.

To obtain f_o_ = 62.41 kHz, a grounded capacitor of 510 pF and a grounded resistance of 5 kΩ were chosen in Figure 2a,b. Figure 13 and Figure 14 show the measured gain and phase frequency responses of the first two proposed circuits for the unity-gain of the non-inverting VMAPF and the inverting VMAPF in Figure 2a and Figure 2b, respectively. For the non-inverting VMAPF pole frequency with a theoretical phase shift of 90°, the measured pole frequency in Figure 13 is 65.29 kHz with an offset of 2.88 kHz. For the inverting VMAPF pole frequency with a theoretical phase shift of −90°, the measured pole frequency in Figure 14 is 65.09 kHz with an offset of 2.68 kHz. Figure 15 and Figure 16 show the measured time-domain input/output waveforms of the non-inverting and inverting VMAPF circuits of Figure 2a and Figure 2b, respectively. At the pole frequency shown in Figure 15, the theoretical phase shift of the first-order non-inverting VMAPF is 90°; the measured phase shift is 89.81°, and the phase error is −0.19°. At the pole frequency shown in Figure 16, the theoretical phase shift of the first-order inverting VMAPF is −90°; the measured phase shift is −88.15°, and the phase error is 1.85°.

A single grounded capacitor of 510 pF and three equal grounded resistors of 5 kΩ and the last two proposed variable-gain control non-inverting and inverting VMAPF circuits shown in Figure 3a,b, are designed for f_o_ = 62.41 kHz. To verify the last two proposed variable-gain control non-inverting and inverting VMAPF circuits in Figure 3a,b, Figure 17 and Figure 18 show the frequency domain measurements of the non-inverting and inverting VMAPFs with an ideal pole frequency of 62.41 kHz, respectively. For a non-inverting VMAPF pole frequency with a theoretical phase shift of 90°, the measured pole frequency in Figure 17 is 62.22 kHz with an offset of −0.19 kHz. For an inverting VMAPF pole frequency with a theoretical phase shift of −90°, the measured pole frequency in Figure 18 is 63.78 kHz, and the offset is 1.37 kHz. Figure 19 and Figure 20 show input/output waveforms in the time domain for the non-inverting and inverting VMAPF circuits of Figure 3a and Figure 3b, respectively. At the pole frequency shown in Figure 19, the theoretical phase shift of the first-order non-inverting VMAPF is 90°; the measured phase shift is 88.48°, and its phase error is −1.52°. At the pole frequency shown in Figure 20, the theoretical phase shift of the first-order inverting VMAPF is −90°; the measured phase shift is −93.26°, and its phase error is −3.26°.

To evaluate the THD, SFDR, IMD3, TOI, PN, and P1dB analysis of the proposed VMAPFs, the proposed paper investigated linearity, dynamic range, harmonic content, mixed, intermodulation linearity, phase noise analysis, and input power range. Figure 21 and Figure 22 show the frequency spectrum of Figure 2a and Figure 2b, respectively. In Figure 21, the calculated THD and measured SFDR values are 0.23% and 57.73 dB, respectively. In Figure 22, the calculated THD and measured SFDR values are 0.29% and 55.59 dB, respectively. Figure 23 and Figure 24 show the THD analysis measured by the operating frequency of 62.41 kHz and the varying input voltage in Figure 2a and Figure 2b, respectively. As shown in Figure 23 and Figure 24, the measured THD is less than 1.13% when the input voltage signal reaches 3.5 V_pp_. Figure 25 and Figure 26 show the frequency spectrum of Figure 3a and Figure 3b, respectively. In Figure 25, the calculated THD and measured SFDR values are 0.5% and 49.81 dB, respectively. In Figure 26, the calculated THD and measured SFDR values are 0.67% and 46.7 dB, respectively. Figure 27 and Figure 28 show the measured THD analysis at an operating frequency of 62.41 kHz versus the varying input voltage in Figure 3a and Figure 3b, respectively. As shown in Figure 27 and Figure 28, the measured THD is less than 1.6% when the input voltage signal reaches 3.5 V_pp_. Figure 29, Figure 30, Figure 31 and Figure 32 show the intermodulation linearity of the two-tone tests with f_1_ = 61.41 kHz for the low-frequency tone and f_2_ = 63.41 kHz for the high-frequency tone, respectively. In Figure 29, the measured IMD3 and TOI of Figure 2a are −57.42 dBc and 33.18 dBm, respectively. In Figure 30, the measured IMD3 and TOI of Figure 2b are −63.91 dBc and 35.18 dBm, respectively. In Figure 31, the measured IMD3 and TOI of Figure 3a are −65.69 dBc and 36.45 dBm, respectively. In Figure 32, the measured IMD3 and TOI of Figure 3b are −52.12 dBc and 31.1 dBm, respectively. Figure 33, Figure 34, Figure 35 and Figure 36 show the PN analysis of approximately 11.6 dBm input carrier power. In Figure 33, the PN measured of Figure 2a at an offset of 100 Hz is −88.61 dBc/Hz. In Figure 34, the PN measured of Figure 2b at an offset of 100 Hz is −89.07 dBc/Hz. In Figure 35, the PN measured of Figure 3a at an offset of 100 Hz is −84.38 dBc/Hz. In Figure 36, the PN measured of Figure 3b at an offset of 100 Hz is −82.68 dBc/Hz. Figure 37, Figure 38, Figure 39 and Figure 40 show the input power range for P1dB, respectively. In Figure 37, the input P1dB of Figure 2a is 19.6 dBm. In Figure 38, the input P1dB of Figure 2b is 19.4 dBm. In Figure 39, the input P1dB of Figure 3a is 19 dBm. In Figure 40, the input P1dB of Figure 3b is 19.6 dBm.

To evaluate the performance of the proposed VMAPF, the figure of merit (FoM) is defined as [42]
(22)$$\mathrm{FoM}\;=\frac{\mathrm{Dynamic}\;\mathrm{Range}\times f_{o}}{\mathrm{Power}\;\mathrm{Disspation}\times \;\mathrm{Supply}\;\mathrm{Voltage}}$$

According to Equation (22), the FoM of the VMAPFs presented in Figure 2a, Figure 2b, Figure 3a, and Figure 3b are 628.2 × 10^3^, 603.06 × 10^3^, 516.53 × 10^3^, and 496.42 × 10^3^, respectively. The summary performance of the proposed VMAPFs is given in Table 2.

To confirm the cascading characteristics, an application example of FQSO in Figure 5 was also studied. The FQSO in Figure 5 was designed with equal capacitor of value 510 pF, R_1_ = R_2_ = R_4_ = R_5_ = R_6_ = 5 kΩ, and R_3_ = 6.2 kΩ. Figure 41 shows the time domain measurements of the quadrature sinusoidal outputs V_o1_ and V_o2_, and the measured oscillation frequency is 59.05 kHz, which is close to the theoretical value of 62.41 kHz. The Fourier spectrum of the quadrature outputs is shown in Figure 42a and Figure 42b, respectively. In Figure 42a, the calculated THD and measured SFDR of V_o1_ output value are 0.8% and 42.88 dB, respectively. In Figure 42b, the calculated THD and measured SFDR of V_o2_ output value are 0.5% and 45.6 dB, respectively. The PN measured of FQSO is shown in Figure 43a and Figure 43b, respectively. At an offset of 100 Hz, the measured V_o1_ and V_o2_ PN are −32.32 dBc/Hz and −32.98 dBc/Hz, respectively. Figure 44 shows the experimental results of the oscillation frequencies of Figure 5 by varying the values of R_1_ = R_4_ with the equal capacitor of value 510 pF, R_2_ = R_5_ = R_6_ = 5 kΩ, and R_3_ = R_6_ = 6.2 kΩ.

## 5. Comparison of VMAPF Theoretical, Simulation, and Experimental Results

To demonstrate the theoretical study of the four VMAPFs, the theoretical analysis of Equations (10), (14), (15) and (17) was performed using Matlab version R2014a software. To evaluate the feasibility of the four VMAPFs, Figure 2a,b and Figure 3a,b were simulated using Cadence OrCAD PSpice version 17.2. To confirm the utility of the four VMAPFs, measurements of the four VMAPFs were performed in Figure 11. Figure 45 and Figure 46 show the theoretical analysis in Matlab and the simulated and measured responses of the first two proposed non-inverting and inverting VMAPF circuits, respectively. Figure 47 and Figure 48 show the simulation and measurement responses of the last two proposed variable-gain control non-inverting and inverting VMAPF circuits, as well as the theoretical analysis in Matlab. In order to illustrate the adjustable features of circuits in Figure 3a,b, R_3_ and R_6_ were changed to 3.15 kΩ, 6.3 kΩ, 10 kΩ, and 15.8 kΩ to obtain −4 dB, 2 dB, 6 dB, and 10 dB corresponding gains, while maintaining C_1_ = C_2_ = 510 pF and R_1_ = R_2_ = R_4_ = R_5_ = 5kΩ. Figure 49 and Figure 50 show the theoretical analysis in Matlab and the simulated and measured variable-gain control without affecting the phase responses in Figure 3a and Figure 3b, respectively. In order to illustrate the adjustable characteristics of the structure displayed, R_1_ in Figure 3a and R_4_ in Figure 3b were changed to 2.5 kΩ, 4 kΩ, 7.5 kΩ, and 15 kΩ to obtain 124.82 kHz, 78.43 kHz, 41.6 kHz, and 20.8 kHz corresponding to f_o_, while maintaining C_1_ = C_2_ = 510 pF and R_2_ = R_3_ = R_5_ = R_6_ = 5 kΩ. Figure 51 and Figure 52 show the theoretical analysis in Matlab and the simulated and measured tunable phase response without affecting the gain in Figure 3a and Figure 3b, respectively. These results show that each structure proposed in Figure 3a,b has variable-gain control without affecting the pole frequency. The experimental and simulation results show that Figure 45 and Figure 52 are in good agreement with the predicted theory, confirming the feasibility of the four proposed VMAPF configurations. However, the differences among the theoretical, simulated, and measured results of the four proposed VMAPFs mainly come from the parasitic impedance effects of active components, passive component tolerances, and PCB circuit layout. To evaluate the difference in the phase shift results of the active and passive components, the sensitivity of the pole angular frequency ω_o_ and gain k parameters of the VMAPF with respect to the passive components of R and C were analyzed using the sensitivity definitions. The sensitivity is defined as [28]
(23)$$S_{x}^{F}=\frac{x}{F}\frac{\partial F}{\partial x}$$

In Equation (23), F represents one of the pole angular frequencies ω_o_; the gain is k, and x represents the passive component of resistor R or capacitor C. Based on Equation (23), the low passive sensitivity of the four VMAPFs is calculated as
(24)$$\text{Non-inverting unity-gain VMAPF}:\;S_{R_{1}}^{\omega_{o}}=S_{C_{1}}^{\omega_{o}}=-1$$
(25)$$\text{Inverting unity-gain VMAPF}:\;S_{R_{4}}^{\omega_{o}}=S_{C_{2}}^{\omega_{o}}=-1$$
(26)$$\text{Non-inverting variable-gain VMAPF}:\;S_{R_{1}}^{\omega_{o}}=S_{C_{1}}^{\omega_{o}}=-1,\;S_{R_{2}}^{k}=-1,\;S_{R_{3}}^{k}=1$$
(27)$$\text{Inverting variable-gain VMAP}:\;S_{R_{4}}^{\omega_{o}}=S_{C_{2}}^{\omega_{o}}=-1,\;S_{R_{5}}^{k}=-1,\;S_{R_{6}}^{k}=1$$

According to Equations (24)–(27), the four proposed VMAPFs have a low passive sensitivity.

## 6. Conclusions

In 2022, Abaci et al. proposed two VMAPFs based on single-modified DDCC−. The first proposed VMAPF uses a modified DDCC−, two grounded capacitors, and a floating resistor, while the second one uses a grounded capacitor, a grounded resistor, and a floating resistor. Both proposed VMAPFs have HI impedance, but they require passive component matching conditions to achieve the VMAPFs and do not offer variable-gain control and LI impedance. In this paper, four new designs for a first-order VMAPF circuit based on two DVCCs are presented. The four proposed VMAPF circuits offer the following attractive features simultaneously: (i) Only GC and GR are used to absorb shunt parasitic capacitances and resistances. (ii) The HI impedance is easily cascaded with other VM circuits without the need for an input voltage buffer. (iii) Inverting APF and non-inverting APF functions do not require matching conditions for passive components. In addition, the first two proposed VMAPFs with LO impedance are beneficial for output cascading, and the latter two proposed VMAPFs with variable-gain control are beneficial for filter-gain controllability. Furthermore, the application as FQSO by cascading the inverting APF and non-inverting APF techniques is discussed. The measured input 1-dB compression point of the four VMAPFs is about 19 dB. The measured THD of the four VMAPFs is less than 0.67% when the input voltage reaches 2.5 V_pp_. The calculated figures of merit for the four VMAPFs are 628.2 × 10^3^, 603.06 × 10^3^, 516.53 × 10^3^, and 496.42 × 10^3^, respectively. The simulation and measurement results are consistent with theoretical predictions.

## Figures and Tables

**Figure 1 sensors-22-06250-f001:**
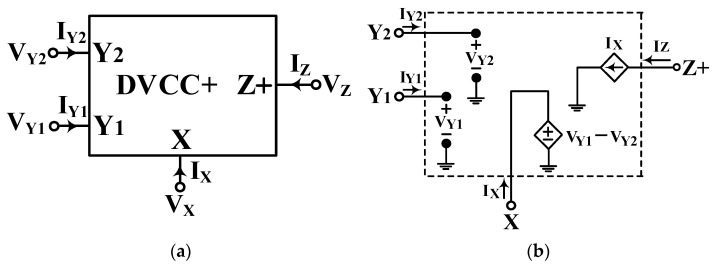
(**a**) Circuit symbol and (**b**) Equivalent circuit of DVCC+.

**Figure 2 sensors-22-06250-f002:**
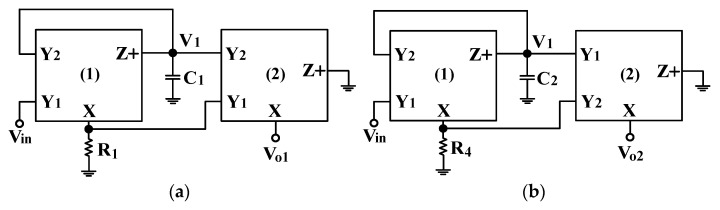
Proposed VMAPFs with HI and LO impedances. (**a**) Non-inverting APF circuit. (**b**) Inverting APF circuit.

**Figure 3 sensors-22-06250-f003:**
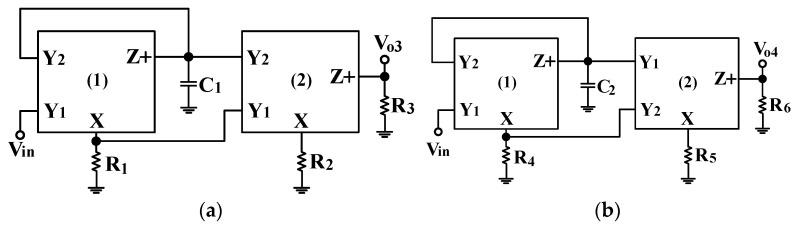
Proposed VMAPF with HI impedance and variable-gain control. (**a**) Non-inverting APF circuit. (**b**) Inverting APF circuit.

**Figure 4 sensors-22-06250-f004:**
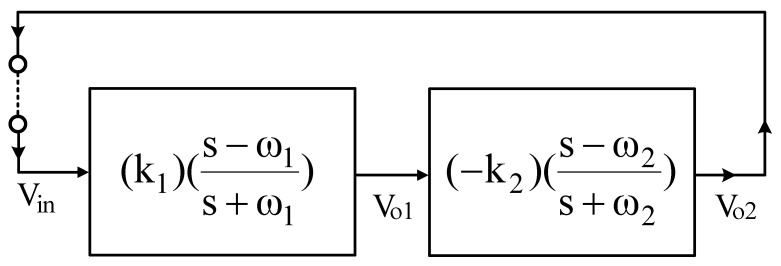
Block diagram of a FQSO by cascading a variable-gain control non-inverting APF transfer function and an inverting APF transfer function.

**Figure 5 sensors-22-06250-f005:**
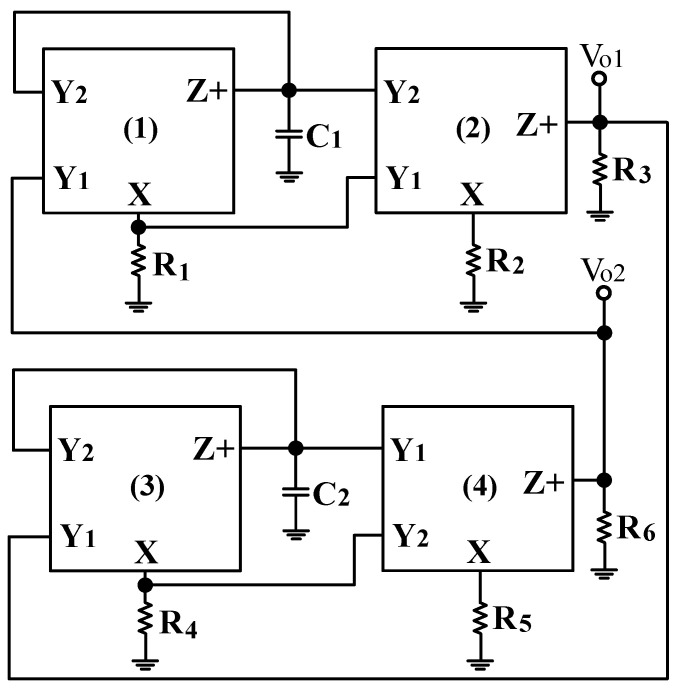
The quadrature sinusoidal oscillator based on cascading a non-inverting APF and an inverting APF with variable-gain control.

**Figure 6 sensors-22-06250-f006:**
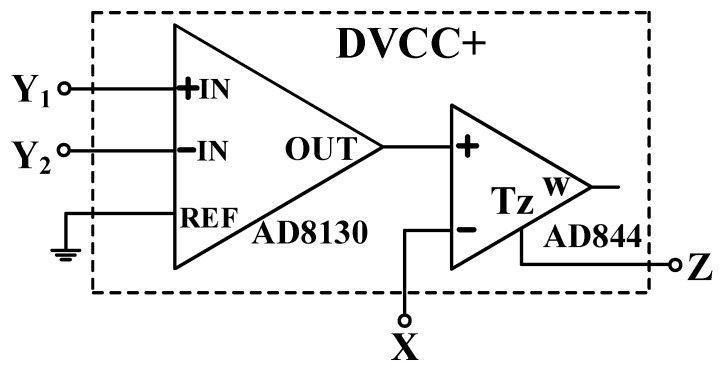
A possible implementation of the DVCC+ using AD8130 and AD844 ICs.

**Figure 7 sensors-22-06250-f007:**
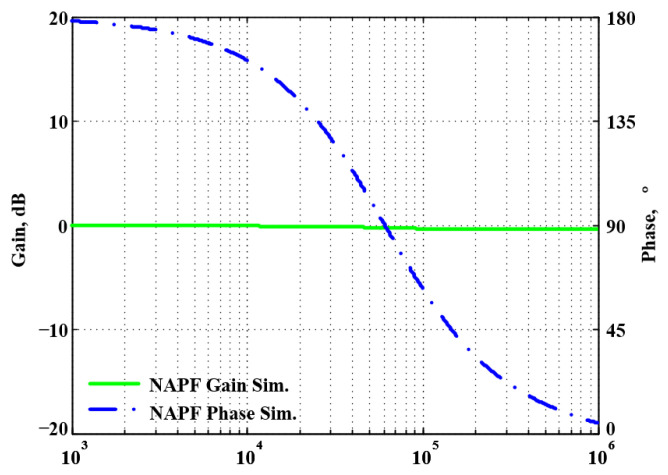
Simulated gain and phase frequency response of the non-inverting VMAPF of Figure 2a.

**Figure 8 sensors-22-06250-f008:**
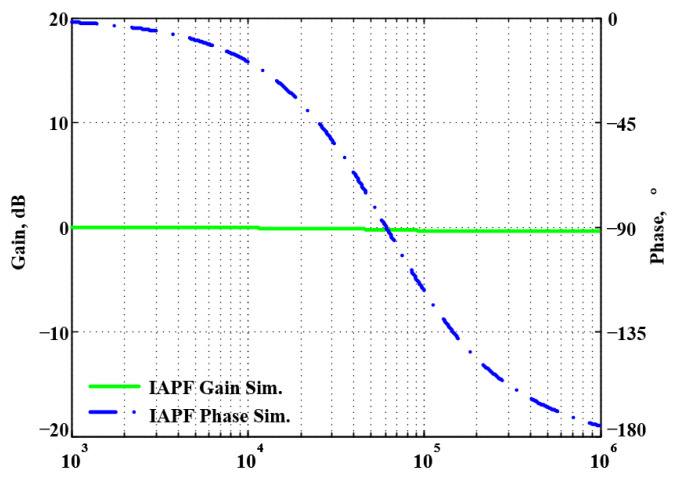
Simulated gain and phase frequency response of the inverting VMAPF of Figure 2b.

**Figure 9 sensors-22-06250-f009:**
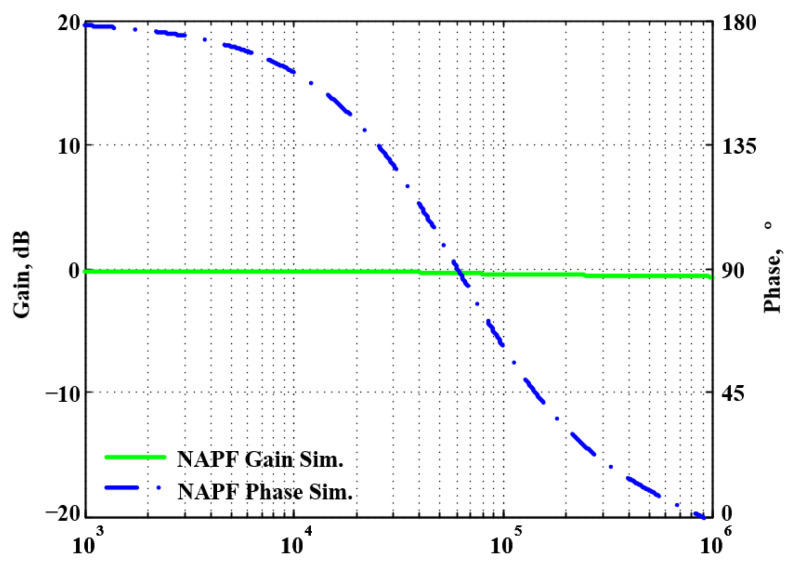
Simulated gain and phase frequency response of the variable-gain control non-inverting VMAPF of Figure 3a.

**Figure 10 sensors-22-06250-f010:**
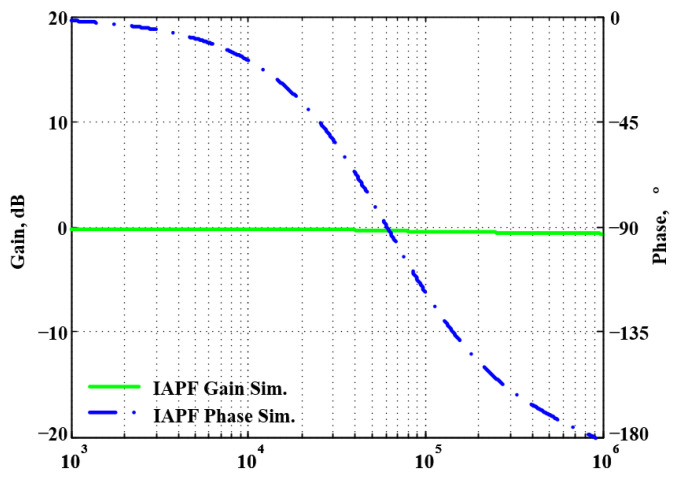
Simulated gain and phase frequency response of the variable-gain control inverting VMAPF of Figure 3b.

**Figure 11 sensors-22-06250-f011:**
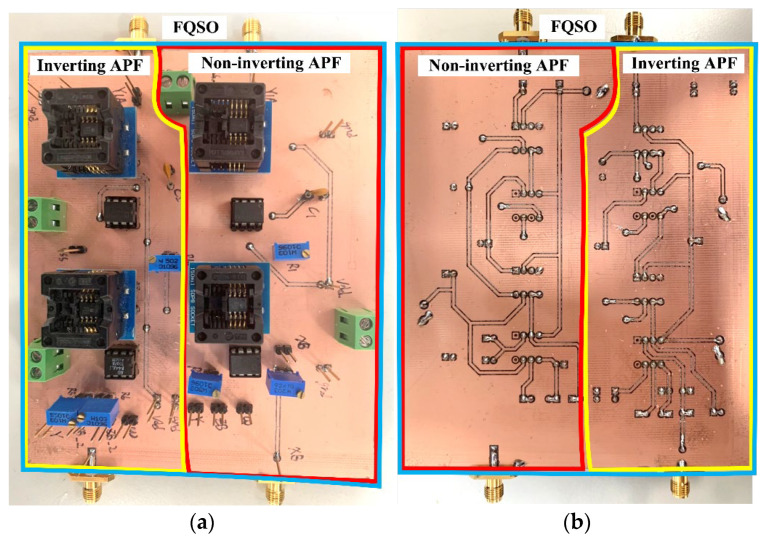
Photographs of the PCB hardware (**a**) top view and (**b**) bottom view for the non-inverting and inverting first-order VMAPFs or second-order FQSO.

**Figure 12 sensors-22-06250-f012:**
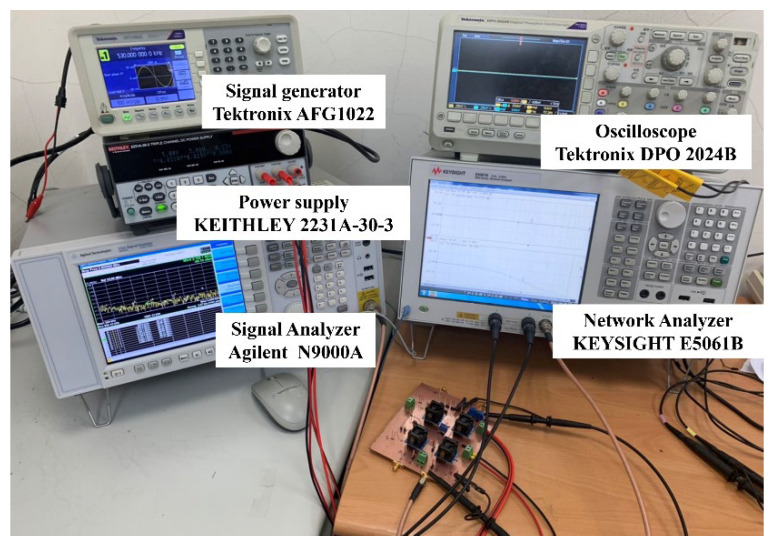
The photograph of the hardware setup used to verify the performance of the proposed non-inverting VMAPF.

**Figure 13 sensors-22-06250-f013:**
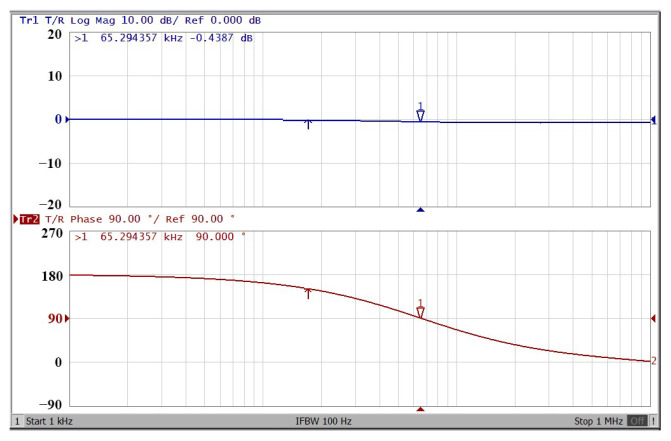
The frequency domain of the non-inverting VMAPF in Figure 2a with the starting frequency range from 1 kHz to 1 MHz.

**Figure 14 sensors-22-06250-f014:**
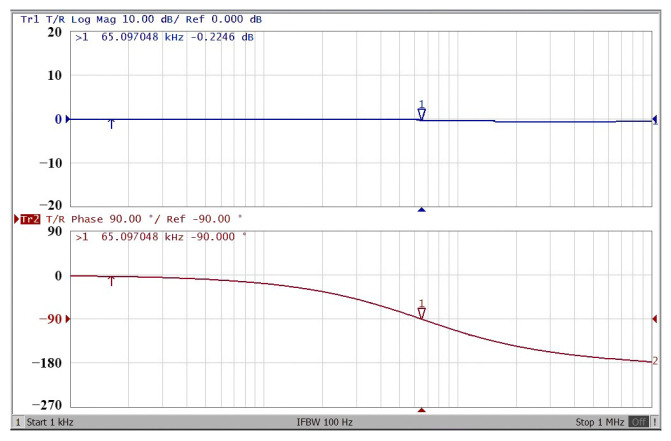
The frequency domain of the inverting VMAPF in Figure 2b with the starting frequency range from 1 kHz to 1 MHz.

**Figure 15 sensors-22-06250-f015:**
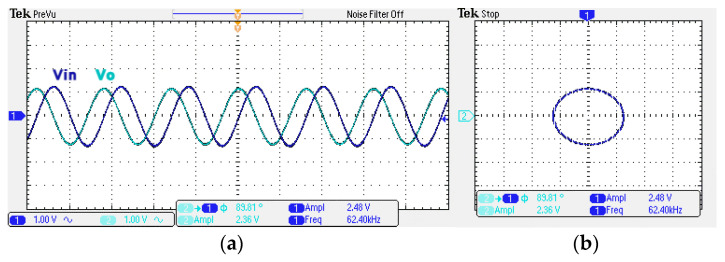
The time domain of the non-inverting VMAPF in Figure 2a with an input signal of 2.5 V_pp_ at 62.41 kHz. (**a**) Input/output signals. (**b**) X–Y plot.

**Figure 16 sensors-22-06250-f016:**
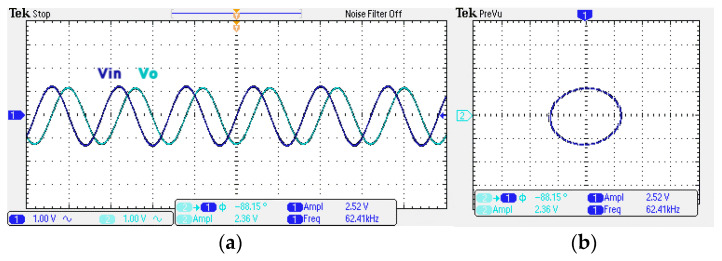
The time domain of the inverting VMAPF in Figure 2b with an input signal of 2.5 V_pp_ at 62.41 kHz. (**a**) Input/output signals. (**b**) X–Y plot.

**Figure 17 sensors-22-06250-f017:**
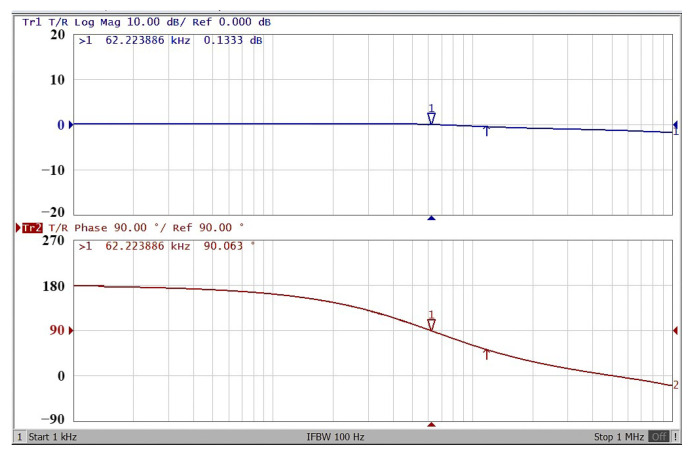
The frequency domain of the non-inverting VMAPF in Figure 3a with the starting frequency range from 1 kHz to 1 MHz.

**Figure 18 sensors-22-06250-f018:**
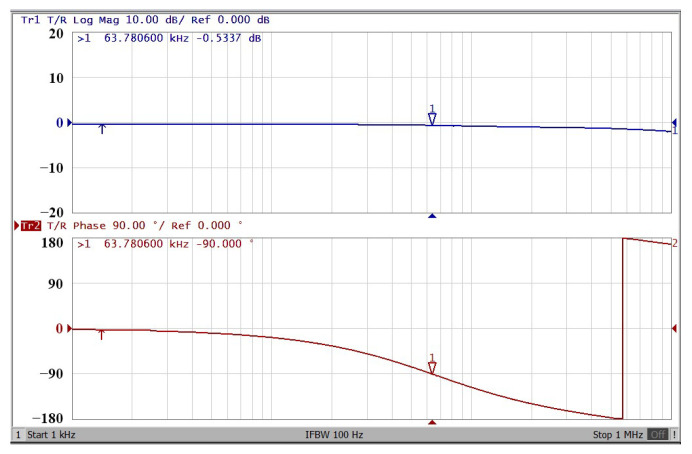
The frequency domain of the inverting VMAPF in Figure 3b with the starting frequency range from 1 kHz to 1 MHz.

**Figure 19 sensors-22-06250-f019:**
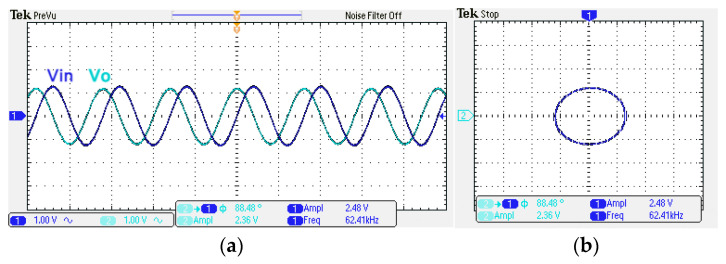
The time domain of the non-inverting VMAPF in Figure 3a with an input signal of 2.5 V_pp_ at 62.41 kHz. (**a**) Input/output signals. (**b**) X–Y plot.

**Figure 20 sensors-22-06250-f020:**
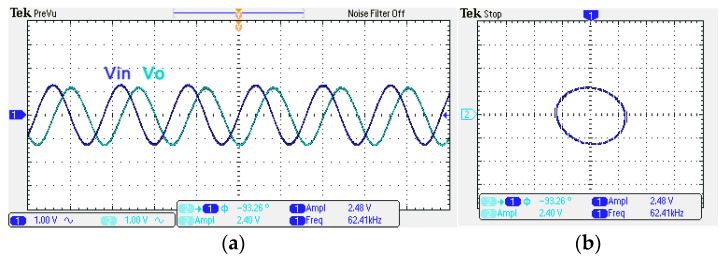
The time domain of the inverting VMAPF in Figure 3b with an input signal of 2.5 V_pp_ at 62.41 kHz. (**a**) Input/output signals. (**b**) X–Y plot.

**Figure 21 sensors-22-06250-f021:**
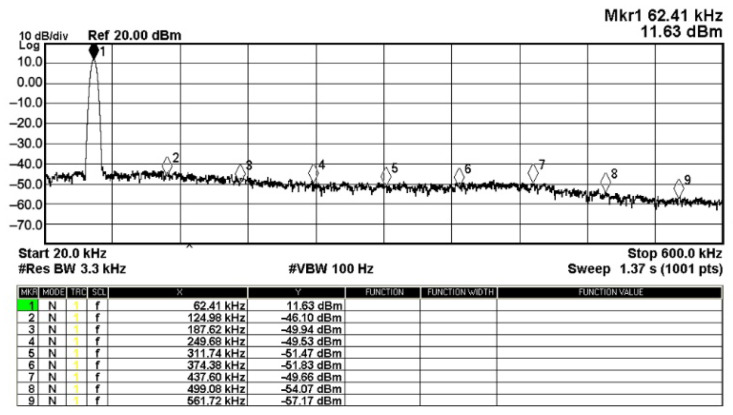
Fourier spectrum of Figure 2a with an input signal of 2.5 V_pp_ at 62.41 kHz.

**Figure 22 sensors-22-06250-f022:**
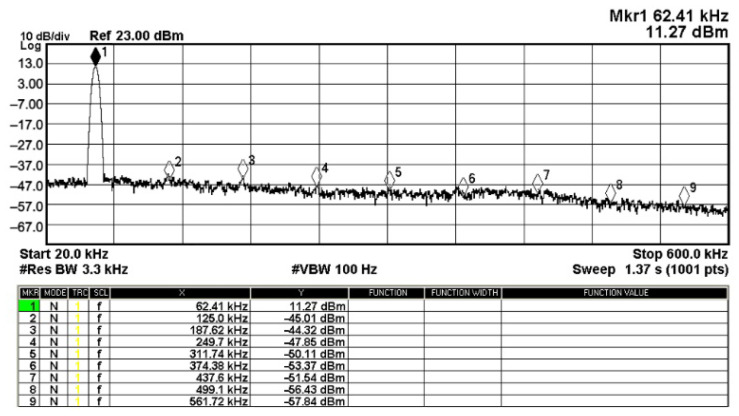
Fourier spectrum of Figure 2b with an input signal of 2.5 V_pp_ at 62.41 kHz.

**Figure 23 sensors-22-06250-f023:**
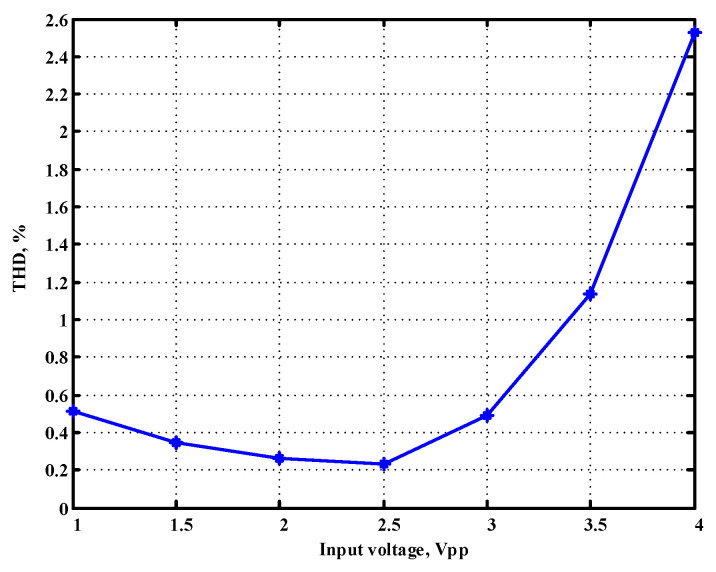
THD measured analysis versus the varying input voltages applied in Figure 2a.

**Figure 24 sensors-22-06250-f024:**
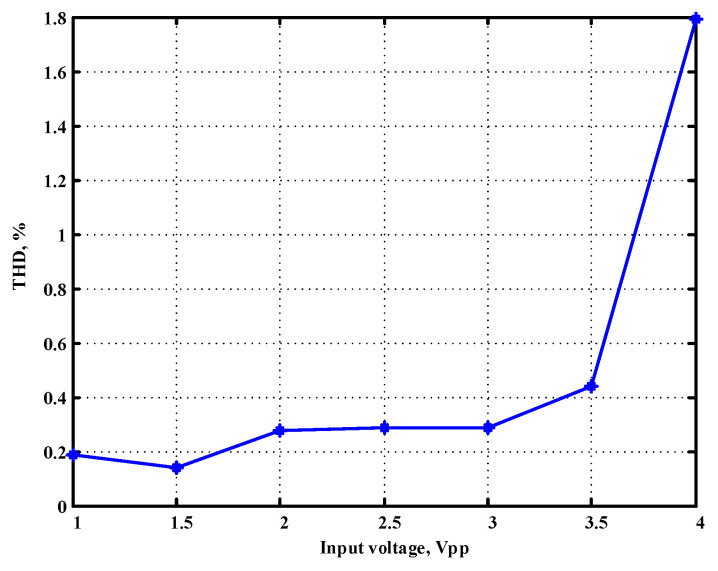
THD measured analysis versus the varying input voltages applied in Figure 2b.

**Figure 25 sensors-22-06250-f025:**
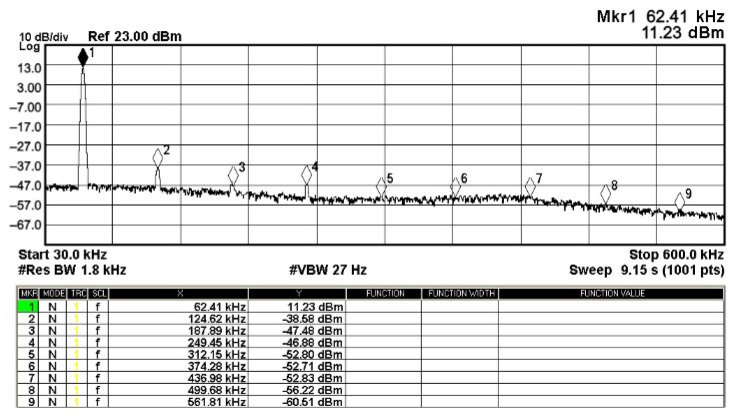
Fourier spectrum of Figure 3a with an input signal of 2.5 V_pp_ at 62.41 kHz.

**Figure 26 sensors-22-06250-f026:**
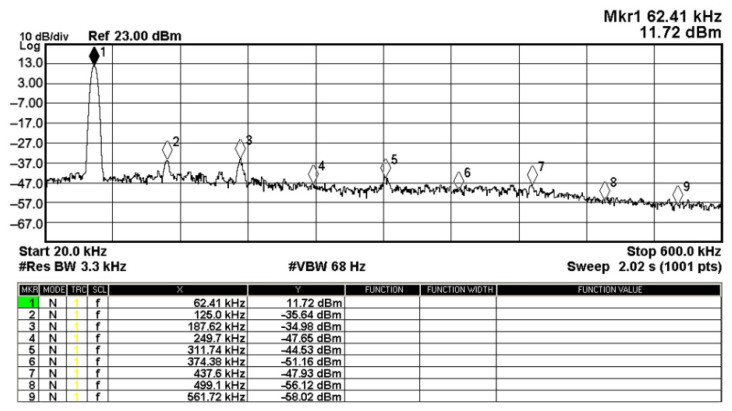
Fourier spectrum of Figure 3b with an input signal of 2.5 V_pp_ at 62.41 kHz.

**Figure 27 sensors-22-06250-f027:**
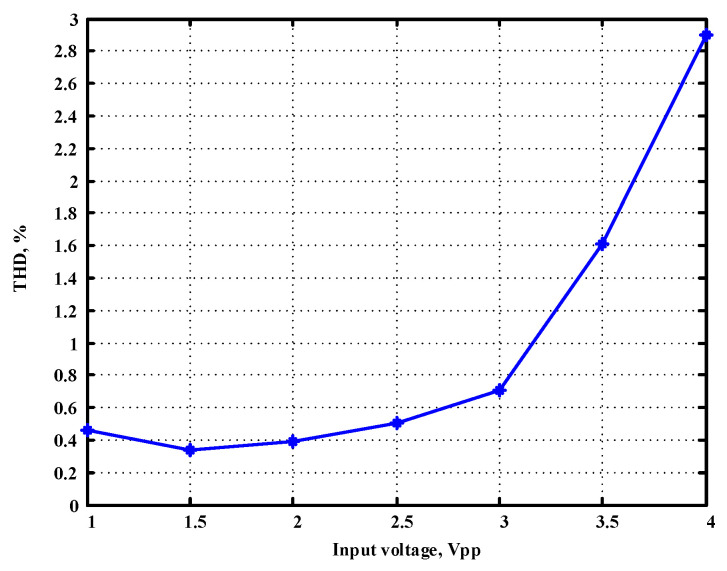
THD measured analysis versus the varying input voltages applied in Figure 3a.

**Figure 28 sensors-22-06250-f028:**
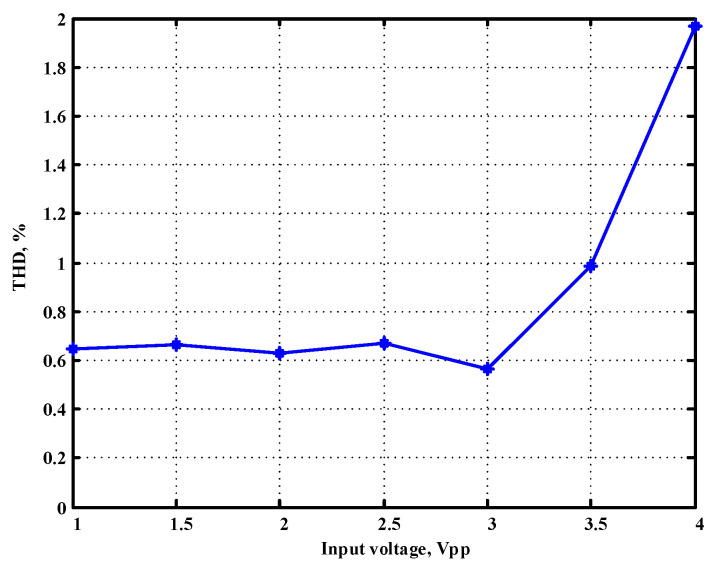
THD measured analysis versus the varying input voltages applied in Figure 3b.

**Figure 29 sensors-22-06250-f029:**
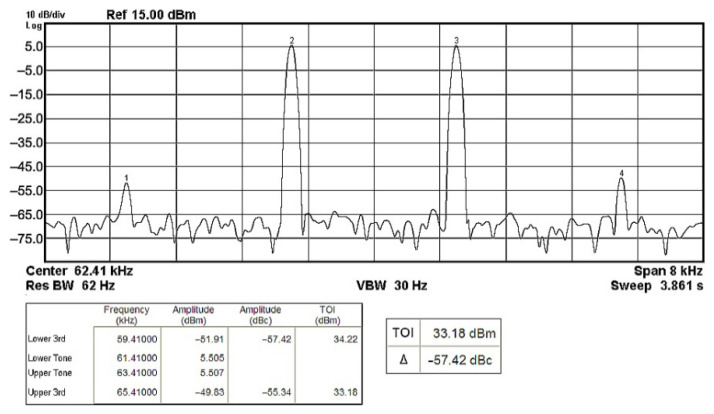
Output spectrum of the two-tone intermodulation distortion test of Figure 2a.

**Figure 30 sensors-22-06250-f030:**
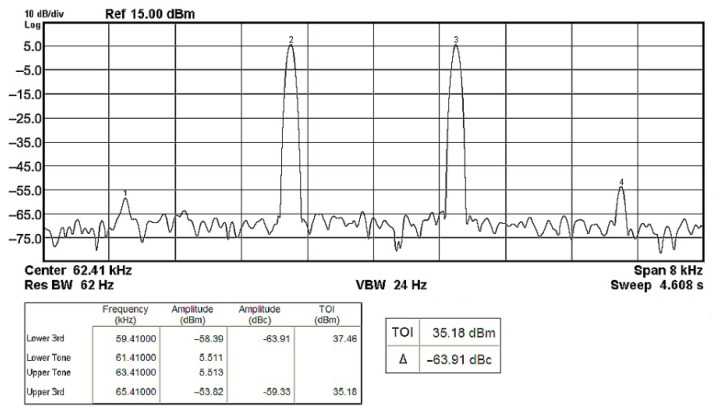
Output spectrum of the two-tone intermodulation distortion test of Figure 2b.

**Figure 31 sensors-22-06250-f031:**
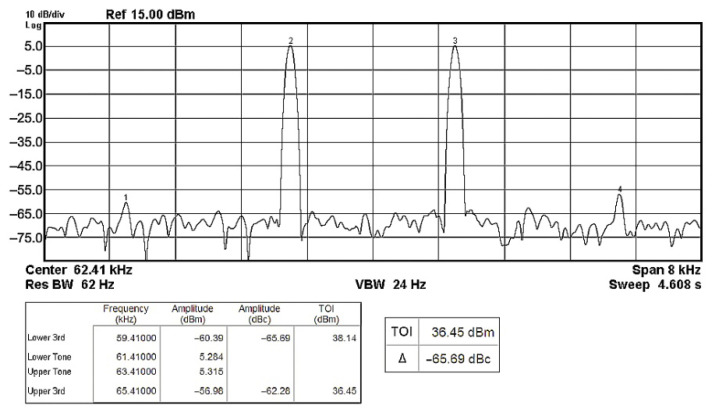
Output spectrum of the two-tone intermodulation distortion test of Figure 3a.

**Figure 32 sensors-22-06250-f032:**
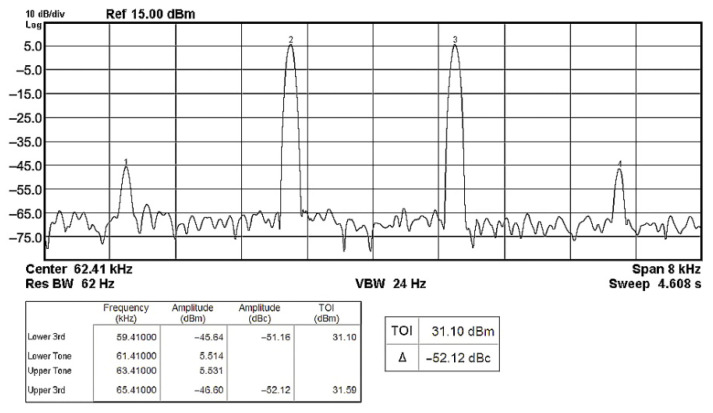
Output spectrum of the two-tone intermodulation distortion test of Figure 3b.

**Figure 33 sensors-22-06250-f033:**
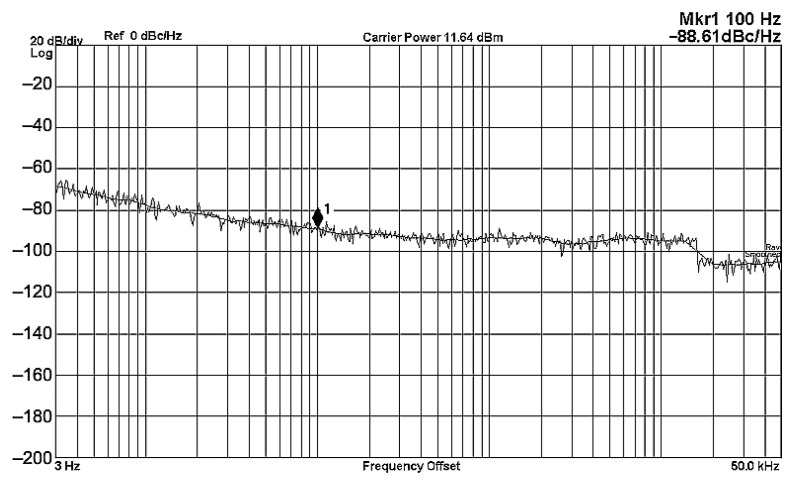
PN analysis of Figure 2a at approximately 11.64 dBm input carrier power.

**Figure 34 sensors-22-06250-f034:**
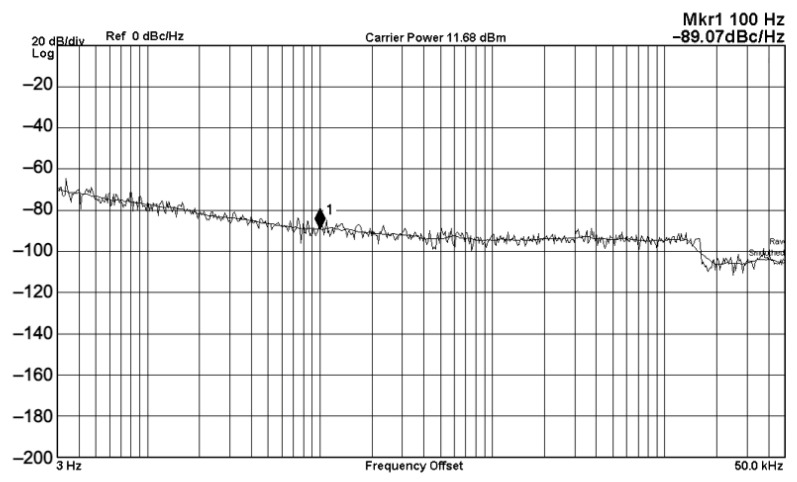
PN analysis of Figure 2b at approximately 11.68 dBm input carrier power.

**Figure 35 sensors-22-06250-f035:**
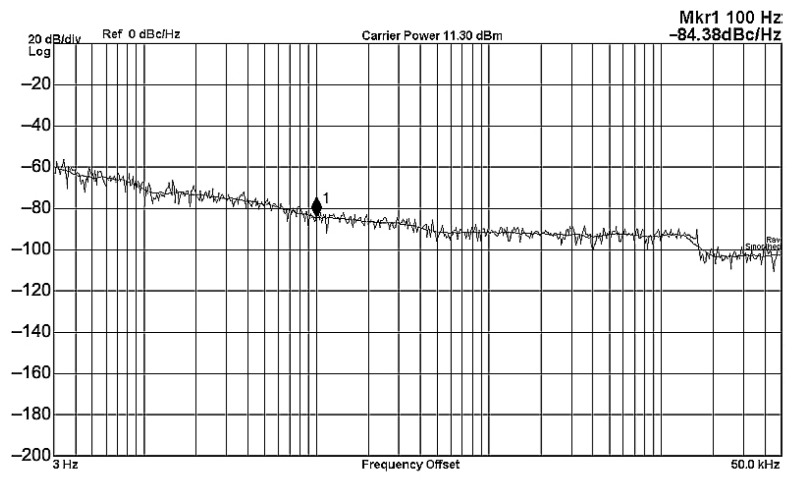
PN analysis of Figure 3a at approximately 11.3 dBm input carrier power.

**Figure 36 sensors-22-06250-f036:**
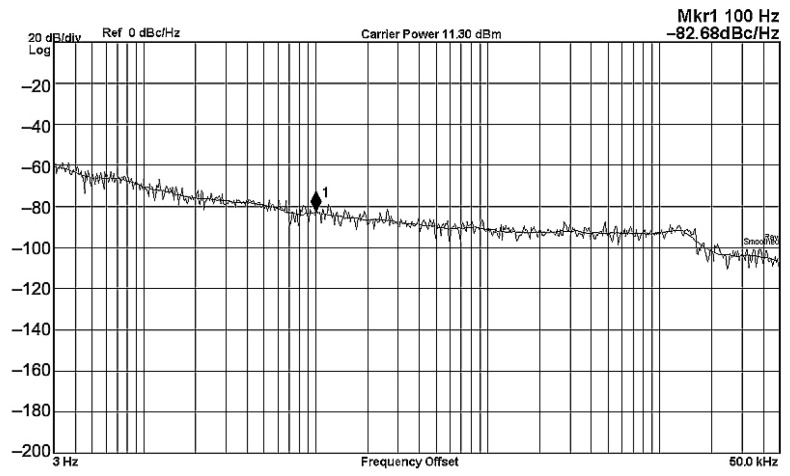
PN analysis of Figure 3b at approximately 11.3 dBm input carrier power.

**Figure 37 sensors-22-06250-f037:**
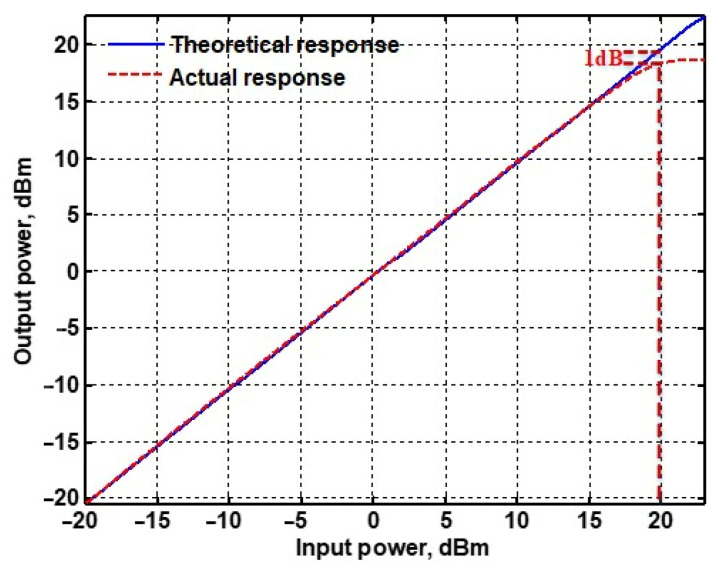
P1dB analysis of Figure 2a.

**Figure 38 sensors-22-06250-f038:**
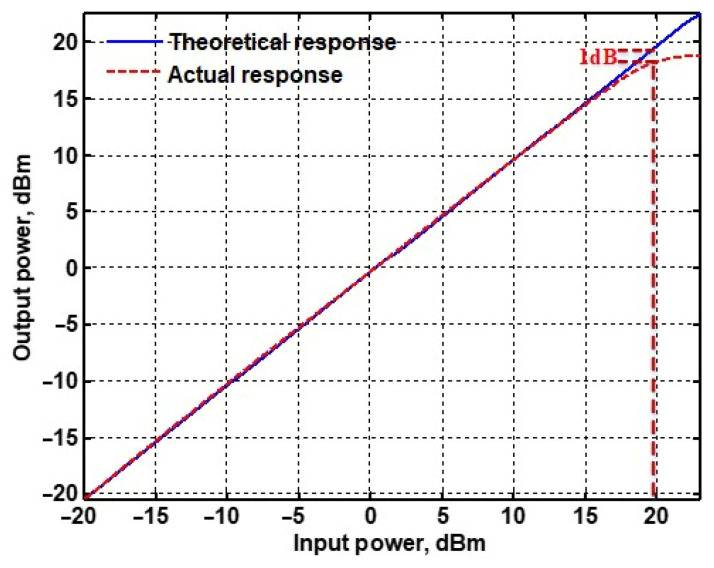
P1dB analysis of Figure 2b.

**Figure 39 sensors-22-06250-f039:**
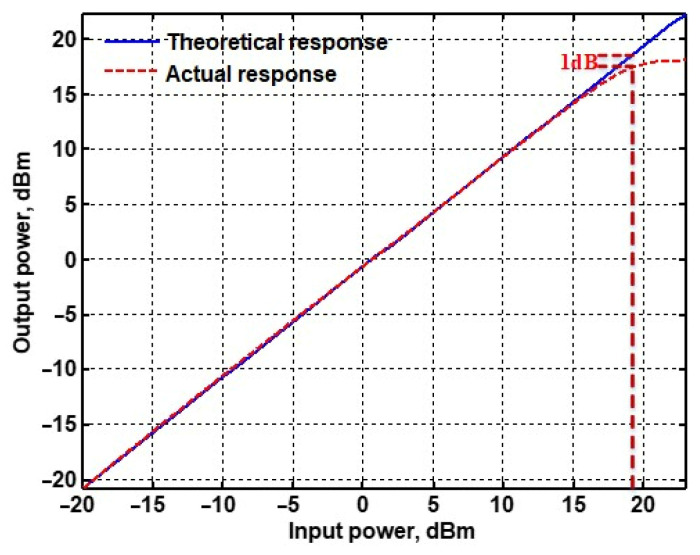
P1dB analysis of Figure 3a.

**Figure 40 sensors-22-06250-f040:**
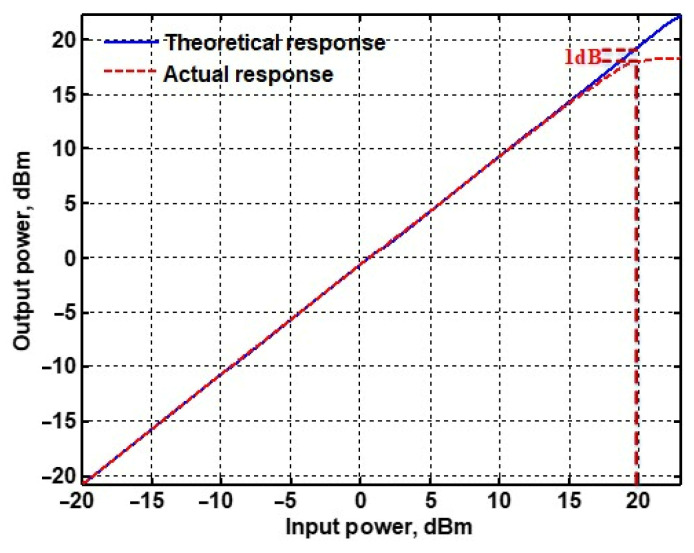
P1dB analysis of Figure 3b.

**Figure 41 sensors-22-06250-f041:**
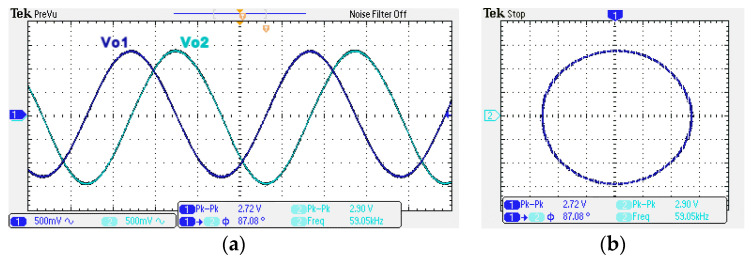
Time domain of FQSO. (**a**) quadrature sinusoidal outputs V_o1_ and V_o2_. (**b**) X–Y plot.

**Figure 42 sensors-22-06250-f042:**
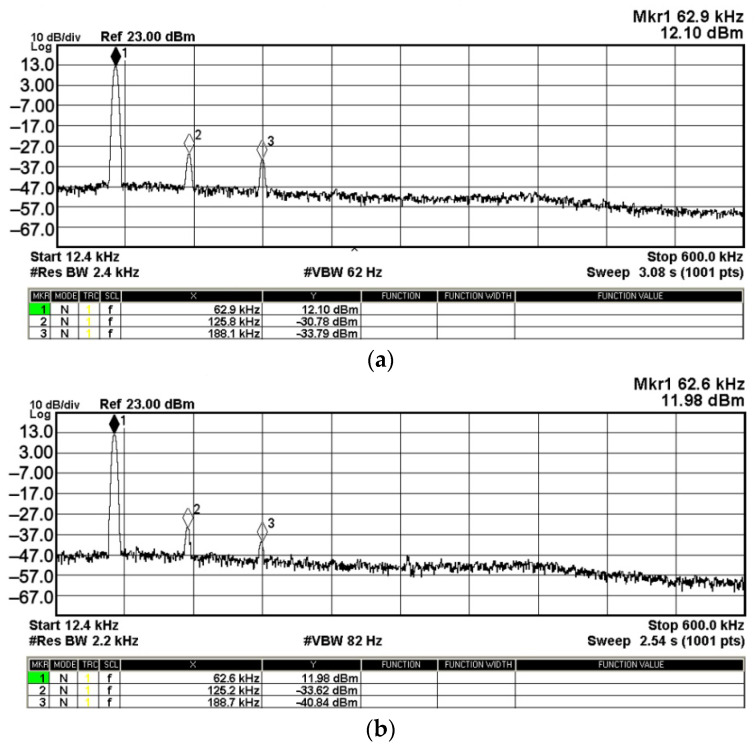
Fourier spectrum of quadrature sinusoidal output signals. (**a**) V_o1_ output spectrum. (**b**) V_o2_ output spectrum.

**Figure 43 sensors-22-06250-f043:**
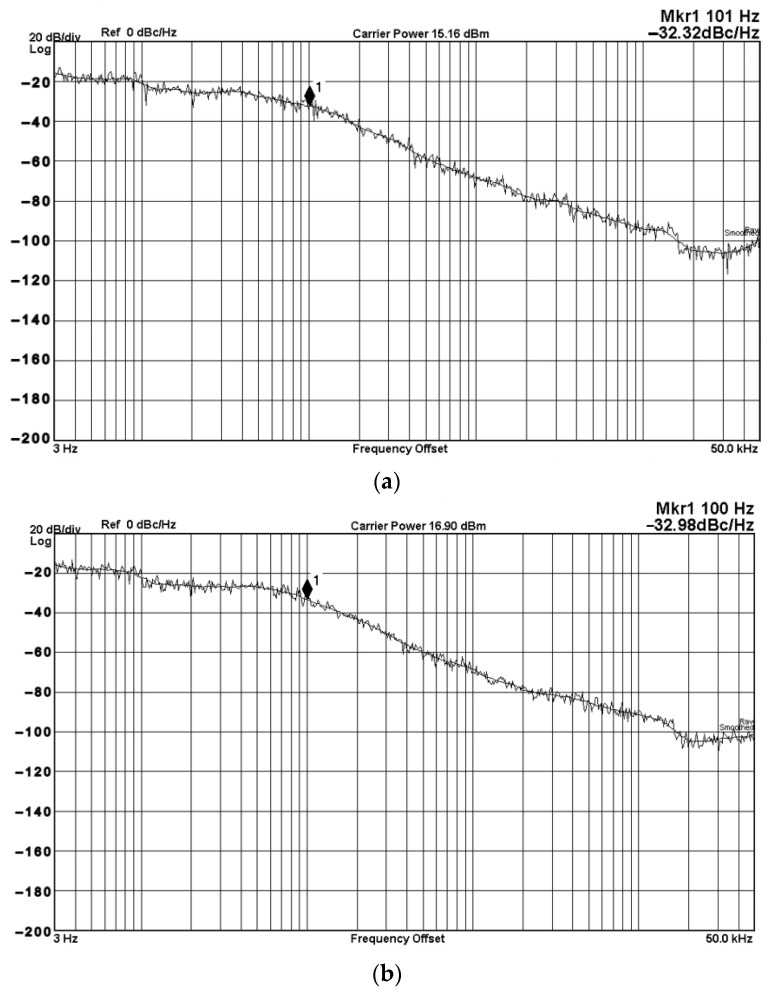
The measured PN of the proposed FQSO. (**a**) V_o1_ output terminal. (**b**) V_o2_ output terminal.

**Figure 44 sensors-22-06250-f044:**
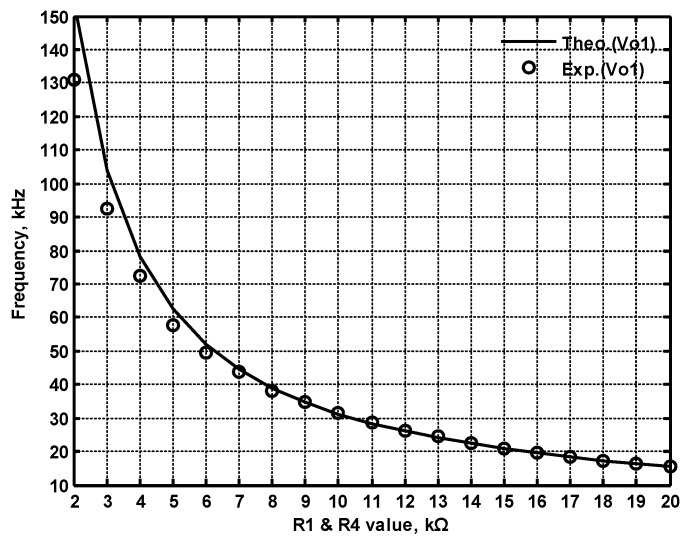
The experimental results of the oscillation frequencies of Figure 5 by varying the values of R_1_ = R_4_ with C_1_ = C_2_ = 510 pF, R_2_ = R_5_ = R_6_ = 5 kΩ, and R_3_ = 6.2 kΩ.

**Figure 45 sensors-22-06250-f045:**
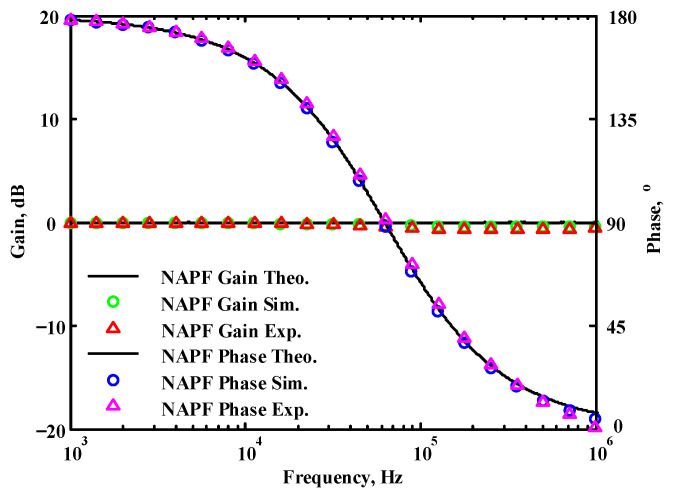
The theoretical analysis in Matlab and the simulated and measured responses of Figure 2a.

**Figure 46 sensors-22-06250-f046:**
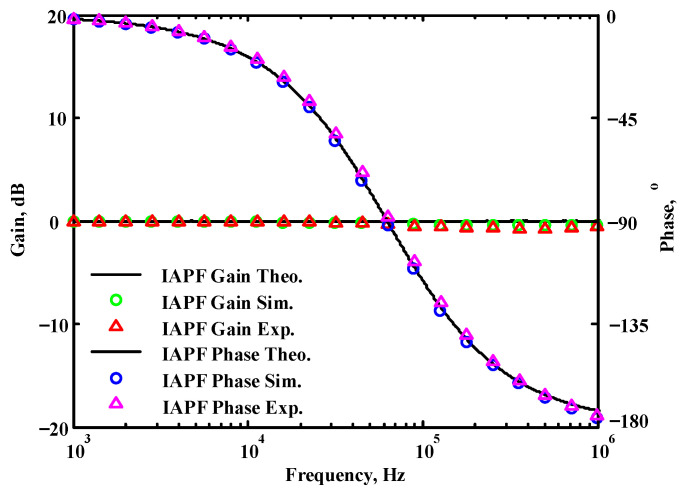
The theoretical analysis in Matlab and the simulated and measured responses of Figure 2b.

**Figure 47 sensors-22-06250-f047:**
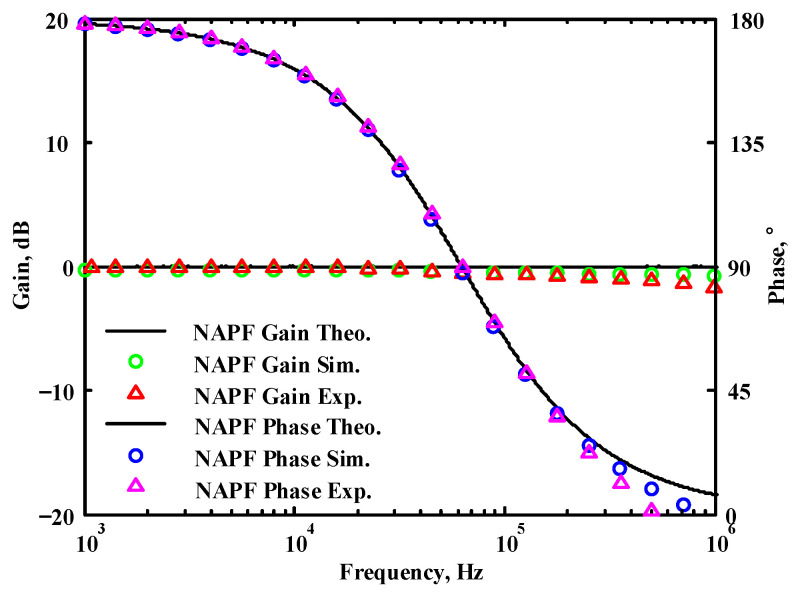
The theoretical analysis in Matlab and the simulated and measured responses of Figure 3a.

**Figure 48 sensors-22-06250-f048:**
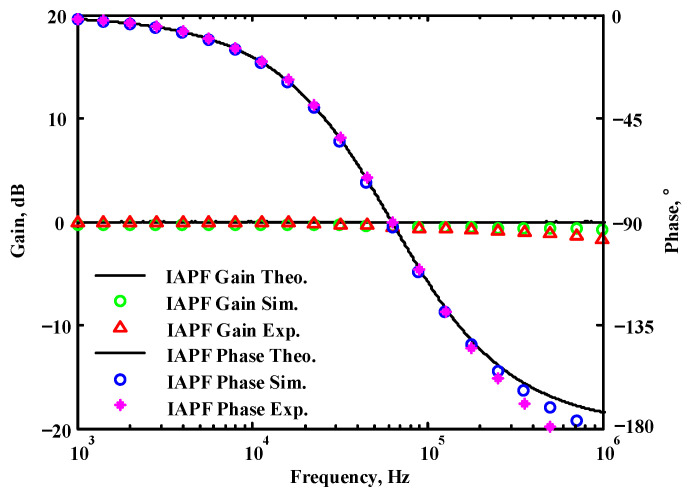
The theoretical analysis in Matlab and the simulated and measured responses of Figure 3b.

**Figure 49 sensors-22-06250-f049:**
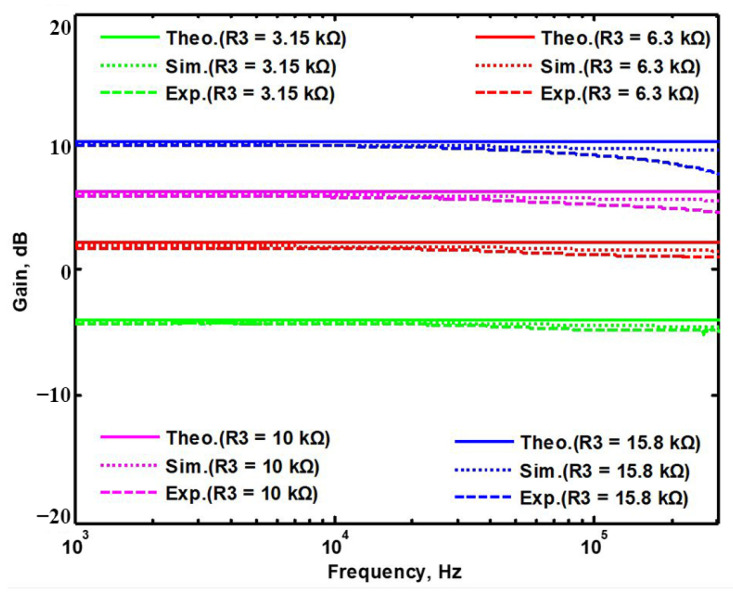
Gain tunability of Figure 3a.

**Figure 50 sensors-22-06250-f050:**
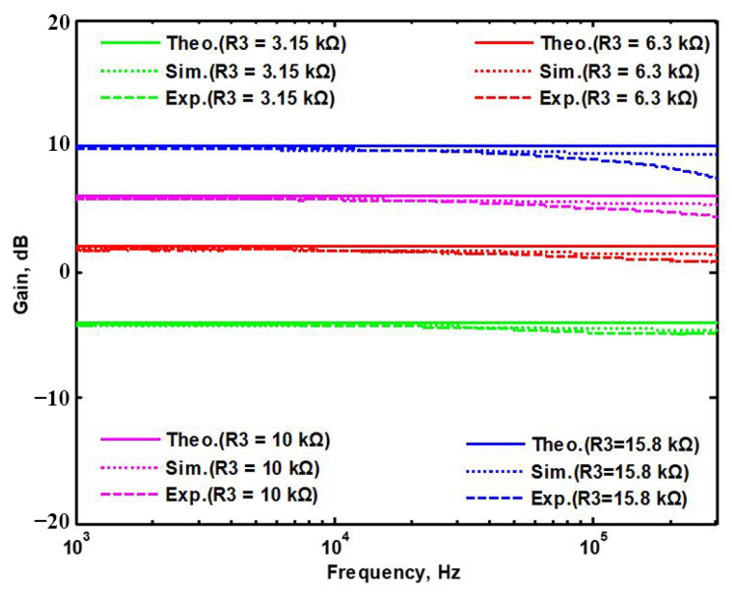
Gain tunability of Figure 3b.

**Figure 51 sensors-22-06250-f051:**
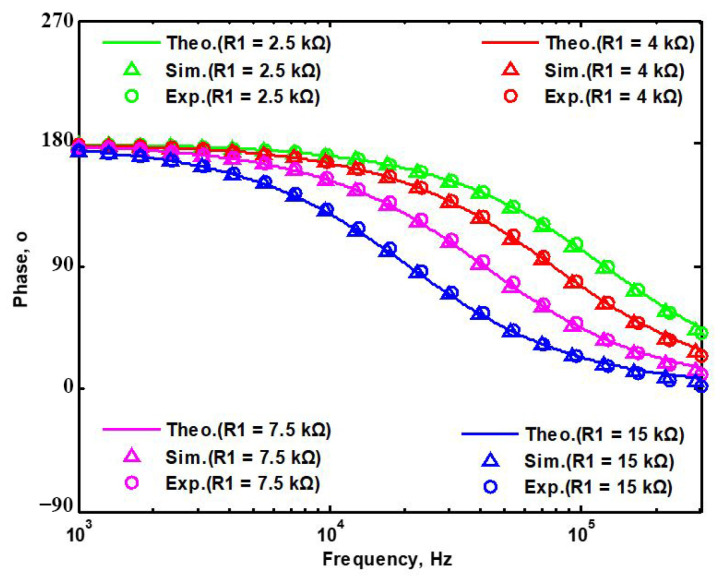
Tunability of pole frequency of Figure 3a.

**Figure 52 sensors-22-06250-f052:**
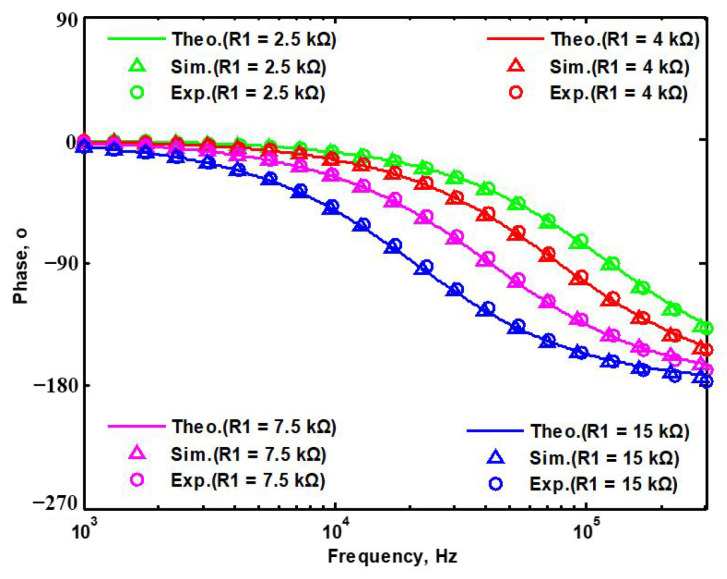
Tunability of pole frequency of Figure 3b.

**Table 1 sensors-22-06250-t001:** An overview of the proposed circuits compared to previously reported VMAPF circuits.

Reference	Use of Active and Passive Elements	Use of Only Grounded Passive Elements	Unlimited Passive Matching Conditions	HI/LO Impedance	Variable-Gain Control	Synthesis of FQSO without Adding Different Structures	Simul./Meas.	Supply Voltage	Power Dissipation
[7]	1 CCII−, 2R, 1C	no	no	no/no	no	no	yes/no	NA	NA
[8]	1 CCII+, 2R, 1C	no	no	no/no	no	no	yes/yes	±12 V	NA
[9]	1 CCII−, 2R, 1C	no	no	no/no	no	no	yes/yes	±12 V	NA
[10]	1 CCII−, 2R, 1C	no	no	no/no	no	no	yes/yes	±12 V	NA
[11]	1 DO-CCII, 2R, 1C	no	no	yes/no	no	no	yes/no	±1.5 V	NA
Figure 3 in [12]	1 DO-CCII, 2 R, 1 C	no	no	yes/no	no	no	yes/no	±0.75 V	Simul. 1.8 mW
[13]	1 UVC, 2R, 1C	no	no	yes/yes	no	yes	yes/yes	±2.5 V	Simul. 5.84 mW
[14]	1 UVC, 2R, 1C	yes	no	yes/yes	no	yes	yes/yes	±1.65 V	NA
[15]	1 ICCII, 2R, 1C	no	no	yes/no	no	no	yes/yes	±2.5 V	NA
[16]	1 DCCII, 2R, 1C	no	no	no/no	no	no	yes/yes	±2.5 V	Simul. 14.3 mW
[17]	1 DXCCII, 3R, 1C	yes	no	yes/no	yes	no	yes/no	±1.25 V	Simul. 2.1 mW
[18]	1 DXCCII, 2R, 1C	no	no	yes/no	no	yes	yes/no	±1.25 V	Simul. 1.8 mW
[19]	1 CDBA, 3R, 1C	no	no	no/no	no	no	yes/no	±2.5 V	NA
[20]	1 EX-CCII,2R, 1C	no	no	yes/no	no	no	yes/no	NA	NA
[21]	1 CCII+, 1 CCII−, 4R, 1C	no	no	no/no	no	no	yes/no	NA	NA
[22]	2 CCII+, 2R, 2C	yes	no	yes/no	no	no	no/yes	±12 V	NA
[23]	2 DVB, 1R, 1C	no	yes	no/yes	no	no	yes/yes	±0.75 V	Simul. 1.77 mW
[24]	2 OTA, 1R, 1C	yes	no	yes/no	no	no	yes/yes	±0.4 V	Simul. 47.2 µW
[25]	1 FDCCII, 1R, 1C	yes	yes	yes/no	no	yes	yes/no	±1.3 V	NA
[26]	1 FDCCII, 1R, 1C	yes	yes	yes/yes	no	yes	yes/no	±3 V	NA
[27]	1 DDCC, 2R, 1C	no	no	yes/no	yes	no	yes/no	±0.9 V	NA
Figure 1 in [28]	1 DDCC, 3R, 1C	no	yes	no/no	yes	no	yes/yes	±1.25 V	NA
Figure 3 in [28]	2 DDCC, 3R, 1C	yes	yes	yes/no	yes	no	yes/no	±1.25 V	NA
[29]	1 DDCC, 1R, 1C	no	yes	no/no	no	no	yes/no	±1.3 V	NA
[30]	1 DDCC, 1R, 1C	no	yes	no/no	no	no	yes/no	±1.25 V	NA
[31]	1 DDCC, 1R, 1C	no	yes	no/yes	no	no	yes/no	±3.3 V	NA
Figure 1a in [32]	1 DDCC, 1R, 1C	no	yes	no/no	no	no	yes/no	±1.5 V	NA
Figure 1b in [32]	2 DDCC, 1R, 1C	yes	yes	yes/no	no	no	yes/no	±1.5 V	NA
Figure 2 in [33]	2 CFOA, 5R, 1C	no	no	no/yes	yes	no	yes/yes	±10 V	0.26 W
Figure 3 in [33]	3 CFOA, 5R, 1C	no	no	yes/yes	yes	no	yes/yes	±10 V	0.39 W
[34]	2 CFOA, 3R, 1C	no	no	yes/yes	yes	no	yes/yes	±8 V	NA
[35]	1 LT1228	no	no	no/yes	yes	no	yes/yes	±5 V	57.6 mW
[36]	2 DVCC, 2R, 1C	yes	no	yes/no	no	no	yes/no	±2.5 V	NA
[37]	2 DVCC, 2R, 1C	yes	no	yes/no	no	no	yes/no	±2.5 V	NA
[38]	2 DVCC, 1Rx, 1C	yes	yes	no/yes	no	no	yes/no	±2.5 V	NA
[39]	2 DVCC, 1R, 1C	no	yes	yes/yes	no	no	yes/no	±1.5 V	Simul. 0.3 W
[40]	2 DVCC, 1R, 1C	yes	yes	yes/no	no	no	yes/no	±1.25 V	NA
Figure 2 in [41]	1 DDCC, 2C, 1R	no	no	yes/no	no	no	yes/yes	±1.25 V	Simul. 4.2 mW
Figure 3 in [41]	1 DDCC, 1C, 2R	no	no	yes/no	no	no	yes/yes	±1.25 V	Simul. 4.21 mW
First proposed 2a in this work	2 DVCC, 1R, 1C	yes	yes	yes/yes	no	yes	yes/yes	±5 V	0.6 W
First proposed 2b in this work	2 DVCC, 1R, 1C	yes	yes	yes/yes	no	yes	yes/yes	±5 V	0.6 W
Second proposed 3a in this work	2 DVCC, 3R, 1C	yes	yes	yes/no	yes	yes	yes/yes	±5 V	0.6 W
Second proposed 3b in this work	2 DVCC, 3R, 1C	yes	yes	yes/no	yes	yes	yes/yes	±5 V	0.6 W

Note: NA: no answer; Simul.: simulation result; Meas.: measurement result; CCII+/CCII−: positive/negative-type second-generation current conveyor; DO-CCII: dual-output second-generation current conveyor; UVC: universal voltage conveyor; ICCII: inverting second-generation current conveyor; DCCII: differential current conveyor; DXCCII: dual-X second-generation current conveyor; CDBA: current differencing buffered amplifier; DVB: subtractor; OTA: operational transconductance amplifier; FDCCII: fully differential second-generation current conveyor; DDCC: differential difference current conveyor; CFOA: current feedback operational amplifier; DVCC: differential voltage current conveyor; HI: high-input; LO: low-output; R: resistor; Rx: X terminal resistance; C: capacitor.

**Table 2 sensors-22-06250-t002:** Performance parameters of the proposed VMAPFs.

Parameter	Figure 2a in This Work	Figure 2b in This Work	Figure 3a in This Work	Figure 3b in This Work
Power supply	± 5 V	±5 V	±5 V	±5 V
Power dissipation	0.6 W	0.6 W	0.6 W	0.6 W
Pole frequency (@ frequency domain)	65.29 kHz	65.09 kHz	62.22 kHz	63.78 kHz
Phase error (@ time domain)	−0.19°	1.85°	−1.52°	−3.26°
THD (@Vin = 2.5 V_pp_)	0.23%	0.28%	0.5%	0.67%
SFDR	57.73 dB	55.59 dB	49.81 dB	46.7 dB
Input P1dB	19.6 dB	19.4 dB	19 dB	19.6 dB
TOI	33.18 dBm	35.18 dBm	36.45 dBm	31.1 dBm
IMD3	−57.42 dBc	−63.91 dBc	−65.69 dBc	−52.12 dBc
PN (@100 Hz offset)	−88.61 dBc/Hz	−89.07 dBc/Hz	−84.38 dBc/Hz	−82.68 dBc/Hz
FoM	628.2 × 10^3^	603.06 × 10^3^	516.53 × 10^3^	496.42 × 10^3^

## Data Availability

Not applicable.

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
