# Peer review of "Four Unity/Variable Gain First-Order Cascaded Voltage-Mode All-Pass Filters and Their Fully Uncoupled Quadrature Sinusoidal Oscillator Applications"

_sensors, 2022, doi:10.3390/s22166250_

Round 1

Reviewer 1 Report

1. The performance of the proposed designs mainly relies on AD8130, and AD844. For first order all pass filter, what are the performance comparison with voltage buffer plus a RC filter? 

2. For fo=62.41 kHz, which is below 100 kHz, what are the typical applications, which cannot be achieve by voltage buffer plus RC circuit? 

3. The reported performances are mainly in-band, what about stop-bands' performance? 

4. What is the group delay? 

5. For measurement such as P1dB, is it 50 Ohm match? 

6. Please also improve the quality of the snapshots in the manuscript. 

7. Quantitative performance in the abstract helps the reader to understand the achievable performance of the proposed designs.

Author Response

The authors would like to express his gratitude to anonymous reviewers for carefully reviewing the paper, for many thoughtful comments in the original manuscript. The manuscript has been revised and improved according to the suggestions of reviewers. The changes in the revised manuscript are marked in blue. We provide a response letter detailing point-by-point revisions to the manuscript and responses to reviewer comments. Please see the attached response.

Reviewer 2 Report

This manuscript presents four new designs for a first-order voltage-mode (VM) all-pass filter (APF) circuit based on two single-output positive differential voltage current conveyors (DVCC+s). In the reviewer's opinion the paper is quite interesting to be accepted to the Journal. However some two points should be improved for the final version of the article:

1.- If should be interesting to clarify the advantages (and the possible drawbacks) of the proposed  designs, comparing them with previous implementations of the state-of-the art.

2.- Please, complete Conclusion Section, highlighting the advantages (and, of course, the possible drawbacks) of the proposed  designs.

Author Response

(The authors gave the same response as above.)

Reviewer 3 Report

Comments

Electrical filters are important components in the construction of electrical and electronic equipment, which is why their optimal dimensioning is very important. The use of operational amplifiers as integrated circuits in the construction of electrical filters has increased their quality. The main requirement of an electric filter is that the extracted signal is as correct as possible, so the frequency band has a width as small as possible, also the attenuation of the signal is as small as possible. Active filters best meet these requirements, but they are also the most expensive. Electric filters, as a rule, are analyzed as quadripole circuits, that is, the output quantities are expressed as a function of the input ones, using the parameters specific to the electric quadripole. Gyrators are quadripolar circuits where the input impedance is reversed compared to the output, if the load impedance is capacitive the input impedance is inductive and vice versa. This property of the gyrators allows the realization of coils (inductances) with a very good quality factor. By connecting a capacitor to the output of the rotor, the impedance seen from the input of the rotor is inductive. This property is preserved for a relatively large frequency range. Considering these elements, I believe that the theme addressed in the paper by the authors is important and useful for technical applications in the field of electronics and electrotechnics. The method used in the study of the filters designed and made by the authors is correct. From those presented in the resulting paper, the simplifying assumptions accepted for the analytical models and for the numerical simulation were confirmed experimentally. In the analytical study and in the numerical simulation the passive circuit elements (resistors and capacitors) were considered ideal. This approximation justifies the differences between the results obtained analytically, respectively by simulation and those obtained experimentally.

Observations

1.       For the clarity of the paper, it is necessary to specify what the variable s represents in relations 1, ..., 19.

2.       Relation 12 is identical to relation 8, why is relation 12 needed and relation 8 is not used?

3.       Figures 11, 12, 17, 18 show results obtained using the Mathlab program and determined experimentally, and in chapter 3 you specify that you use the PSpice programming environment for the numerical simulation. How is that right?

4.       Differences in phase shift result from the fact that the circuit elements are not ideal but real. Could you improve the mathematical model used to eliminate the differences between the analytical and experimental results?

5.       How can the results shown in figures 27, 28 and 31 be justified, from which it follows that with increasing voltage the total distortion coefficient increases.

6.       6. The results obtained by simulation should be presented in a separate chapter from those obtained experimentally. I propose that the results obtained by simulation should be presented in chapter 3, those obtained experimentally in chapter 4, and in chapter 5 the results obtained analytically and by simulation should be compared with those obtained experimentally. Doing so makes the paper clearer and easier to analyze.

Author Response

(The authors gave the same response as above.)

Reviewer 4 Report

Review of the paper sensors-1862828 : Four Unity/Variable Gain First-Order Cascaded Voltage-Mode All-Pass Filters and Their Fully Uncoupled Quadrature Sinusoidal Oscillator Applications

 The paper brings theoretical and experimental results related to frequency selective circuits and their application.

I have a few comments and suggestions for the authors.

1-     The number of recent references could be increased.

2-     Some figures are centered, and others are not centered.

3-     I suggest the authors change the color of the text in Figure 7 (a) and Figure 8 since it is not easy to see the text.

4-     I suggest the author change the dimension of Figure 11. That is the same for other Figures, when it is possible, like Figures 24, 27, 41, and 42.

5-     At the end of the paper, the authors could present a discussion comparing the obtained results with other authors and approaches for the same problem.

6-     The above comment is also essential for the conclusion since there are a lot of experimental results, and the author did not mention their relevance in the conclusions. The authors could include a field of application for the proposed circuits (and justify it).

t

Author Response

(The authors gave the same response as above.)
